

# Development and characterization of a high-efficiency, aircraft-based axial cyclone cloud water collector

Ewan Crosbie[1,2], Matthew D. Brown[1,3], Michael Shook[1], Luke Ziemba[1], Richard H. Moore[1], Taylor Shingler[1,2], Edward Winstead[1,2], K. Lee Thornhill[1,2], Claire Robinson[1,2], Alexander B. MacDonald[4],
Hossein Dadashazar[4], Armin Sorooshian[4,5], Andreas Beyersdorf[6], Alexis Eugene[7], Jeffrey Collett, Jr.[8], Derek Straub[9], Bruce Anderson[1]

[1]NASA Langley Research Center, Hampton, VA 23666, United States.
[2]Science Systems and Applications, Inc. Hampton, VA 23666, United States.
[3]Universities Space Research Association, Columbia, MD 21046, United States.
[4]Department of Chemical and Environmental Engineering, University of Arizona, Tucson, AZ 85721, United States.
[5]Department of Hydrology and Atmospheric Sciences, University of Arizona, Tucson, AZ 85721, United States.
[6]California State University, San Bernardino, CA 92407, United States.
[7]Department of Chemistry, University of Kentucky, Lexington, KY 40506, United States.
[8]Atmospheric Science Department, Colorado State University, Fort Collins, CO80523, United States.
[9]Department of Earth and Environmental Sciences, Susquehanna University, Selinsgrove, PA 17870, United States.

*Correspondence to*: Ewan Crosbie (ewan.c.crosbie@nasa.gov)

**Abstract.** A new aircraft-mounted probe for collecting samples of cloud water has been designed, fabricated and extensively tested. Following previous designs, the probe uses inertial separation to remove cloud droplets from the airstream, which are

subsequently collected and stored for offline analysis. We report details of the design, operation, and modelled and measured probe performance.

Computational Fluid Dynamics (CFD) was used to understand the flow patterns around the complex interior geometrical features that were optimized to ensure efficient droplet capture. CFD simulations coupled with particle tracking and multiphase surface transport modelling provide detailed estimates of the probe performance across the entire range of flight operating

conditions and sampling scenarios.

Physical operation of the probe was tested on a Lockheed C-130 Hercules (fuselage mounted) and de Havilland Twin Otter (wing pylon mounted) during three airborne field campaigns. During C-130 flights on the final field campaign, the probe reflected the most developed version of the design and a median cloud water collection rate of 4.5 ml min[-1] was achieved. This allowed samples to be collected over 1-2 minutes under optimal cloud conditions. Flights on the Twin Otter featured an inter-

comparison of the new probe with a slotted-rod collector, which has an extensive airborne campaign legacy. Comparison of trace species concentrations showed good agreement between collection techniques, with absolute concentrations of most major ions agreeing within 30%, over a range of several orders of magnitude.



## 1 Introduction

Clouds are a fundamental characteristic of the hydrological cycle and an important component of the Earth's climate system. They affect the transport of pollutants and act as a primary mechanism for the vertical redistribution of trace species (e.g. Pickering et al., 1988, Andreae et al., 2004, Seigel and van den Heever, 2012, Barth et al., 2015, Twohy et al., 2015, Corr et

al., 2016). Clouds drive the primary removal mechanism for atmospheric aerosol particles through precipitation formation (Seinfeld and Pandis, 2016), but also produce changes in aerosol composition through aqueous reactions (e.g. Sorooshian et al., 2007, Ervens et al., 2008, Wonaschuetz et al., 2012). Aqueous reactions are known to play a significant role in the production of secondary aerosol species, including sulfate (Pandis et al., 1992, Collett et al., 1994) and organics (Ervens et al., 2004, Carlton et al., 2009) and in the case of secondary organic aerosols, these processes are underrepresented in models

because of their complexity and the difficulty of obtaining satisfactory validation data (Carlton et al., 2009). Remote sensing of aerosols in the vicinity of clouds is challenging (Koren et al., 2007), and in situ sampling of clouds and regions near clouds requires an aircraft, except in some unique locations such as mountaintop sites (e.g. Collett et al., 2002, Marinoni et al., 2004, Schwab et al., 2016) and in locations encountering fog (e.g. Collett et al., 1998, Herckes et al., 2007, Li et al., 2011, Straub et al., 2012, Weiss-Penzias et al., 2012, Herckes et al., 2013).

Airborne collection of bulk cloud water samples for subsequent chemical analysis has proven to be a useful tool for quantifying the abundance of chemical species within clouds. Other complementary techniques such as the counterflow virtual impactor (Twohy et al., 2003, Shingler et al., 2012) dry the cloud droplets to produce residual particles, which are then analyzed for composition or size, but this generally has the disadvantage of expelling semi-volatile aerosol species and dissolved gases present in the cloud droplets and also presents challenges with recovering absolute concentrations. Although cloud water

sampling for investigating aerosol activation, pollutant transport and chemical evolution now spans several decades, recent measurements have shown skill in promoting linkages between cloud water chemistry and dynamic processes. Examples of this are the characterization of the coupling state of marine clouds with the ocean surface (Wang et al., 2016), the identification of above and below cloud sources (Wang et al., 2016, Braun et al., 2017) and the connection with physical properties, such as the presence of giant cloud condensation nuclei (Dadashazar et al., 2017). In addition, progress has been made with using

cloud water samples to diagnose wet scavenging in marine clouds (MacDonald et al., 2018).

The majority of airborne cloud water samples have been collected using the slotted-rod collector (Mohnen, 1980), the operation of which is widely described (Hegg and Hobbs, 1986, Huebert et al., 1988, Kim and Boatman, 1992, Leaitch et al., 1992). However, this technique is difficult to operate on pressurized aircraft thereby limiting the locations and vertical extent of sampling, and its performance is sensitive to the operating conditions (Huebert et al. 1988), fabrication, and materials (Huebert

and Baumgartner, 1985). Axial cyclones also have former use as inertial separation devices for cloud water (e.g. Walters et al., 1983) and feature a swirl-generating device driven by ram pressure to separate droplets from the airstream. A new type of collector featuring an axial cyclone was developed by Straub and Collett (2004, hereafter SC04) for use on the NCAR C-130



and flight tested in 2001. The collector was used during two field campaigns with airborne deployments over the eastern Pacific to study stratocumulus cloud composition (Straub et al., 2007, Benedict et al., 2012). This probe was wing mounted in a standard probe canister, which resulted in favorable air flow characteristics but the small space for sample storage limited the overall number of samples to seven per flight. The collection of cloud water was also slower and less efficient than expected,

requiring long sample durations (> 10 min) to collect sufficient volume for offline analyses.

In this paper, we describe efforts that have improved the axial cyclone technique through the design of a new probe, which we have named the Axial Cyclone Cloud water Collector (AC3). This was accomplished through multiple design improvements (Sect. 2) and aerodynamic optimization supported by computational fluid dynamics (CFD) simulations (Sect. 3). The new probe performance was evaluated in flight during multiple airborne field campaigns (Sect. 4). Through an operational inter-

comparison with the slotted-rod, the AC3 has shown good agreement with compositional analysis of samples as summarized in Sect. 5. Considerable increases in collection efficiency and sampling rate are achieved with the new design, which is suitable for both pressurized and non-pressurized aircraft. These advances translate directly into improved sample time resolution and sample volumes.

## 2 Design Summary

### 2.1 Specifications

The AC3 concept expands and revises the axial cyclone concept described by SC04 and aims to specifically address limitations in collection efficiency associated with the original design. SC04 provide a thorough review of the operation of axial cyclones, together with a description of computational and physical modelling techniques used to verify performance. Briefly, the axial cyclone technique involves imparting swirl (using a device that we call the "stator") along the direction of the free stream,

creating a helical flow pattern. The resulting helical flow centrifugally separates large particles and droplets from the air stream, since the migration of particles towards the outer periphery of the vortex is dependent on particle size. For optimal airborne cloud water collection, the cloud droplets migrate to the walls for capture, while interstitial aerosol particles are not captured and remain in the air stream and are exhausted. In addition, the design should maximize the volume of collected cloud water to allow the possibility of high temporal resolution sampling and, potentially, online analytical techniques. A

schematic cross section of the current design is shown in Fig. 1 and a summary of key specifications is given in Table 1. The probe parts were machined from aerospace-grade 6061-T6 aluminum with the exception of the stator, which was manufactured using the direct metal laser sintering technique from 316 stainless steel powder. In an attempt to improve the surface properties to minimize contaminants and droplet adhesion, the aluminum parts were later anodized using a Teflon-impregnated hardcoat (Alexandria Metal Finishers, Lorton, VA) and the stator was coated with Super Koropon 515-700 primer (PPG Aerospace,

Valencia, CA). All parts were then dip-coated and cured with a fluoropolymer hydrophobic surface treatment (PFC1300; Cytonix, Beltsville, MD). Samples of all these products were tested in the laboratory prior to their inclusion on the probe to



ensure there were no undesirable artifacts. The laboratory process involved preparing material samples using the same surface treatment as the probe components and then soaking in ultrapure water for 24 hours. The water was then tested for major ions and organic carbon and no statistically significant enhancement was observed compared to untreated samples.

**2.2 Stator**

Swirl is generated in the AC3 through a single-stage stator comprising eight blades. The stator blades present a blockage to the incoming flow and therefore their design affects the total air flow through the probe. For example, the greater the blade curvature and associated angular momentum production, the greater the pressure drop across the stator, reducing the total mass flow. However, maximizing the angular momentum is desirable for efficient separation of droplets from the airstream; thus the geometry of the AC3 stator reflects an optimization of axial and tangential flow components to maximize the water
collection.

Theory and reduced-order modelling were used to determine the optimal blade curvature and stator geometry during the design. The blade cross section was based on the NACA 4-series airfoil with a 12% thickness to chord ratio. The blade camber was implemented by mapping the airfoil profile onto a camber line which was defined by a variable blade trailing edge angle, $\phi$,
while the axial length of the blade, $L_{BLADE}$, was held fixed at 50 mm. For simplicity, the aerodynamics were analyzed as a two-dimensional azimuthal cross-section at the mid-span line (area-based). Geometrical arguments were used to determine a discharge plane at the throat created between the blade trailing edge and the adjacent blade. The throat width, $w(\phi)$, dictates the exit jet between blades and constrained the separation point on the blade upper surface, while the region between adjacent jets was modelled as a zero-momentum wake. The discharge angle, $\beta(\phi)$, of the jet was assumed to be parallel to the camber
line angle at the separation point. An illustration of the geometry and separation points is shown in Fig. 2a. An additional displacement thickness, $\delta^*$, was added to the upper and lower surface of the blade to include boundary layer skin friction losses, using the simplified Blasius form:

$$\delta^* = \frac{1.72x}{\sqrt{Re_x}} \tag{1}$$

where x corresponds to the streamwise distance along the blade, and $Re_x = Ux/\nu$ is the local Reynolds number on the blade with respect to x, U is the local velocity outside the boundary layer and $\nu$ is the momentum diffusivity. Boundary layer effects were found to add only a small contribution for most blade angles. The wake fraction, $f_w$, is defined:

$$f_w = 1 - \frac{w - \delta^*_{TE} - \delta^*_{sep}}{s \cdot cos(\beta)} \tag{2}$$





where w is the throat width determined from the blade geometry, and $\delta_{TE}$ and $\delta_{sep}$ represent the displacement thicknesses at the trailing edge and upper surface separation point, respectively. The wake fraction is equivalent to the azimuthal fraction of a cross section immediately downstream of the blades that comprises the wake, with the remaining fraction comprising the jets. By continuity, and using $\beta$, the axial and tangential components of the jet velocity, $U_2$ and $V_2$, are determined by the inlet
velocity, $U_1$, upstream of the stator and can be written:

$$\frac{U_2}{U_1} = \frac{1}{1-f_w} \tag{3}$$

$$\frac{V_2}{U_1} = \frac{tan(\beta)}{1-f_w} \tag{4}$$

We conservatively assume that no pressure recovery occurs from the downstream dissipation of the swirl flow and there is no cross-stream pressure gradient between jet and wake regions, which yields a loss coefficient, $K_{1\text{-}2}$, for the stator:

$$K_{1-2} = \frac{P_1 - P_2}{\frac{1}{2}\rho U_1^2} = [cos(\beta)(1 - f_w)]^{-2} - 1 \tag{5}$$

The inlet velocity for the probe can now be estimated relative to the free stream, $U_0$, based on the stator loss coefficient and combined losses from other probe components, which we assume is dominated by probe exit discharge and the restriction in cross section downstream of the stator, which we denote $K_{2\text{-}3}$ and set equal to 2.1 based on $R_{pipe}$ and R (see Fig. 1).

$$P_{macro} = \frac{U_1}{U_0} = \left(\frac{1}{K_{1-2}+K_{2-3}+1}\right)^{\frac{1}{2}} \tag{6}$$

The axial velocity ratio defined in Eq. (6) has a direct connection to the fraction of cloud droplets that enter the probe. Droplet inertial effects caused by streamline curvature at the sub-isokinetic inlet and the potential for impaction on the stator are an important feature of the probe performance but we consider these aspects later in the detailed simulations and analyses. For
purposes of design, the inlet axial velocity ratio serves as a first-order proxy for the fraction of the available cloud water that enters the probe and we refer to it as the macrophysical performance metric, $P_{macro}$. The other factor affecting performance relates to the microphysically dependent process of capturing droplets within the probe. For this, we need to define a droplet Stokes relaxation time, $\tau$:

$$\tau = \frac{\rho_w D^2}{18\mu} \tag{7}$$





where $\rho_w$ is the density of water, D is the droplet diameter and $\mu$ is the dynamic viscosity of air. The migration of droplets to the outer walls for capture is principally dependent on $U_1$, $\beta$, $\tau$, the capture zone radius, R, and the capture zone length, L:

$$\frac{dr_p}{dt} = \frac{\tau(U_1 tan\beta)^2}{r_p} \tag{8}$$

where $r_p$ is the radial position of a droplet. Integrating Eq. (8) defines a trajectory equation whereby the axial displacement, $\Delta x$, of a droplet defined by $\tau$ can be uniquely determined for the displacement from initial radial position $r_0$ to the outer wall at R:

$$r_0{}^2 - R^2 = \tau U_1 (tan\beta)^2 \Delta x \tag{9}$$

We define $r_{0,min}$ such that droplet collection at any $r_0 > r_{0,min}$ satisfies $\Delta x < L$, so the area fraction ($\chi$) of collectable droplets can be written:

$$\chi = \frac{r_{0,min}{}^2 - R^2}{R^2} = \frac{\tau U_1 (tan\beta)^2 L}{R^2} \tag{10}$$

Since this area fraction has unit upper bound, the use of a single $\tau$ to characterize the droplet microphysics is overly simplistic and the droplet capture is better modelled with a droplet size distribution. We assume a log-normal distribution of droplet diameter with geometric standard deviation, $\sigma$, resulting in a log-normal relaxation time distribution centered on $\tau$ with

geometric standard deviation $2\sigma$ (see Eq. 7) and normalized density function $n(\tau)$. The microphysical performance metric is therefore a weighted integral across all relaxation time satisfying:

$$P_{micro} = \int n(\tau) \, min(\chi(\tau), 1) \, d\tau \tag{11}$$

For optimization, we consider the product of the performance metrics, $P = P_{macro} * P_{micro}$ , as a proxy for the efficiency of the stator with respect to blade trailing edge angles ranging from 0° to 75°. The absolute value of this metric is less meaningful than the variation with respect to blade angle. Fig. 2b shows the analysis performed for a range of microphysical scenarios with droplet mass size distributions centered at 10 μm, 20 μm, and 30 μm diameters, and with $U_0 = 120$ ms$^{-1}$. Increasing the curvature of the blades reduces the flow through the probe, which degrades performance at high blade angles. For smaller

droplets, a moderate-to-high blade angle is needed for successful capture, while this requirement becomes less important for large droplets. The assumptions of the model are less representative for further increases in droplet size since the relaxation time is too long for the equilibrium assumptions to be realistic. We also note that the model makes no attempt to capture the

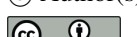

radial profile of the velocity components. Nonetheless, the optimization model provides guidance for picking a blade angle based on the expected range of diameter of cloud droplets (Fig. 2c). A 50° design blade angle was selected to offer a balance of performance across the anticipated range of droplet sizes; however, the blade angle was reduced towards the root to satisfy fabrication and structural limitations.

## 2.3 Extraction

The AC3 features a new approach for extracting the cloud water as compared to SC04. The captured cloud water on the walls is forced rearwards towards an annular extraction chamber as a result of the action of aerodynamic drag associated with the helical flow aft of the stator. The opening of the extraction chamber – the extraction slot – is bounded by the outer, cylindrical wall of the probe and a flared cylindrical pipe through which the exhaust air passes (hereafter, we refer to it simply as the "pipe"). The pipe can be adjusted in the axial direction across ten positions (P1-P10, where P1: L=13 mm, P10: L=76 mm with uniform increments of 7 mm), which allows control over the length of the capture zone and is used for design evaluation. The extraction chamber is maintained at slight negative pressure relative to the main body of the probe by a vacuum pump inside the aircraft, which is intended to help maintain a positive airflow from the main body into the collection chamber. Once inside the extraction chamber, cloud water settles gravitationally to the lowest point where the sample is pumped out through a single port and into storage vials inside the aircraft cabin. Another feature of the vacuum line, not yet tested in flight, is the potential to measure water vapor or other gases, which in concert with diagnostic pressure measurements inside the probe permits real-time monitoring of evaporation losses.

## 2.4 Storage and Metering

Collected cloud water was pumped through a standard 1/8-inch perfluoroalkoxy (PFA) line into the cabin of the aircraft using a constant flow rate peristaltic pump. The pump ran continuously while sampling was taking place and usually the tubing contained a mix of air and water. Analysis of capillary and viscous forces with this tubing internal diameter (1.57 mm) and flow rate (50 ml/min) indicated a laminar slug flow regime (Dukler and Hubbard, 1975, Fabre and Liné, 1992), where water and air phases periodically span the entire tube cross section, permitting optical flow metering. Two non-intrusive, optical bubble sensors (Panasonic Industrial, BE-A301) are arranged in series on the sample tubing to detect the liquid and air fraction during sampling, and hence provide a method of metering the volume of collected cloud water at high temporal resolution. Laboratory testing revealed that across a wide range of simulated water collection rates, slug flow prevailed with the exception that at very low water availability, the bubble sensor tracking became more intermittent, which may complicate tracking water collection in marginal cloud conditions when liquid water content (LWC) is low.

The collected water is either dispensed into sealed Nalgene® storage vials in a programmable auto-collector (Brechtel Manufacturing Inc.) or manually into polypropylene centrifuge tubes that are capped immediately after collection. Data acquisition software logs the collection rates and total volume; if the auto-collector is used, it controls the active vial position.





Samples can be collected over a prescribed volume or duration to allow flexibility to accommodate various offline analysis requirements.

**2.5 Shutter**

While the collector is designed to selectively remove cloud droplets but not aerosol particles from the airstream, there is the
potential for accumulation of particles on internal surfaces during flight outside clouds. Even though the collection of aerosols is designed to be inefficient, the high air flow rates (>10000 LPM) through the probe together with any exposure during an extended period of time produces an unsatisfactory accumulation of surface material leading to a contamination effect on subsequent cloud water collection. When the inlet was continuously open during initial flight testing the contamination issue was found to be particularly apparent in conditions with enhanced coarse mode aerosol particles, such as sea salt or dust, and
in biomass burning plumes. An operable shutter mechanism closes the inlet to air flow when not sampling clouds. The shutter is actuated by a servo motor controlled by the user interface within the data acquisition software and limit switches housed next to the servo motor provide feedback to the software that the shutter is in the desired position. Analysis of pre- and post-flight blanks, with and without the shutter, confirmed the necessity of its use and its successful function for removing aerosol contamination outside of cloud (see Sect. 5).

**3 Design evaluation through multiphase modelling**

**3.1 CFD Simulations**

3.1.1 Model Description

Three separate models were constructed to assess the flow structure around the inlet, the stator and the capture zone, respectively. This approach allows the complex three-dimensional flow around the stator blades to be investigated in isolation
of the simpler, rotationally symmetric probe body. The inlet and capture zone simulations used 2D axisymmetric simulations, which benefit from lower computational cost and can be run iteratively to quickly assess design changes and sensitivities.

Simulations were run using the ANSYS-FLUENT software solving the compressible Reynolds-Averaged Navier Stokes (RANS) governing equations until convergence and were interpreted as a steady-state mean flow solution. The turbulence
scheme was the standard RANS $k$-$\varepsilon$ model. The upstream boundary conditions reflected the range of nominal flight operating conditions during low-altitude (0.5 − 3 km) cloud sampling for a C-130 (115 ms$^{-1}$) or a Twin Otter (50 ms$^{-1}$). Of critical importance to assessing performance across the flight envelope was the degree of flow self-similarity (i.e. whether the streamline pattern is consistent after scaling velocities and/or lengths, in proportion), which allows results generated from one scenario to be scalable to other cases (e.g. C-130 to Twin Otter). By default, we used 115 ms$^{-1}$ at 850 hPa and 5°C (typical of
~1.5 km altitude), but to assess the sensitivity of the self-similarity assumption, we conducted additional simulations on each





domain with conditions representative of the Twin Otter aircraft and found the scaled results to agree within 5%. The simulations were executed in series beginning with the stator simulation, which was run to determine downstream velocity components and pressure ratios, relative to prescribed inlet conditions. These ratios were consolidated into axisymmetric profiles and bulk coefficients that characterized the losses and swirl generating performance of the stator. The bulk coefficients

were added to the inlet simulation, which modelled internal and external flows, primarily to determine the probe inlet velocity ratio relative to the freestream. Finally, detailed capture zone simulations were run for different pipe positions with boundary conditions that reflected the velocity ratio profiles from the stator simulation, and the inlet velocity from the inlet simulation.

3.1.2 Flow Field

The primary purpose of the stator is to generate swirl. Velocity profiles (Fig. 3a) are shown at seven axial positions upstream and downstream of the blades and illustrate the generation of the swirling flow. A region of flow separation is evident at the stator hub behind the blades, which may permit considerable unsteadiness in the flow field (not evident from RANS simulations) and potentially reduce the effectiveness of droplet separation in the wake. The loss coefficient, K, is defined as

the areal mean drop in total pressure, $\Delta P_0$, across the stator normalized by the upstream dynamic pressure, $q_{ref}$.

$$K = \frac{-\langle \Delta P_0 \rangle}{q_{ref}} \qquad (12)$$

where <*> denotes an area mean (across azimuthal and radial dimensions). The swirl number, S, is defined as the ratio of

axial flux of angular momentum to the axial flux of axial momentum:

$$S = \frac{\langle \rho r U_\theta U_x \rangle}{R \langle \rho U_x^2 \rangle} \qquad (13)$$

where $\rho$ is the air density, $U_x$ and $U_\theta$ are axial and tangential velocity components and the radial coordinate, r, extends from the centerline to the outer wall at R. Both K and S accumulate along the stator axis (Fig. 3b) with an approximately linear

generation of swirl across the blade chord. Losses are also generated across the blades, although there is a large fraction of the total loss that arises from the flow separation on the hub. Additional offline sensitivity studies conducted with CFD, suggested that the separation on the hub was more pronounced with strongly swirling flows. The bulk effects of the stator on the interior flow can be represented by the K and S coefficients, subject to the condition of flow self-similarity. Even in the absence of losses (K=0), the generation of swirl creates an effective momentum deficit in the axial direction because of the

static pressure gradient within the vortex, which affects the flow rate within the probe and impacts on performance (see Sect. 2.2). The effective total loss can be written:

$$K_{eff} = K + cS^2 \qquad (14)$$





Where c is a shape factor that accounts for the combined profile shapes of the axial and tangential velocity and compressibility effects (radial density gradient). The $S^2$ dependence holds under the dimensional scaling assumptions of self-similarity and the CFD simulations predicted c = 2.86 for this stator geometry.

Inlet simulations were run to determine the details of flow streamlines surrounding the exterior of the probe and to provide an estimate of the flow rate of the air passing through the probe interior. Changes to the internal geometry or a change in the stator head loss coefficient affects whether streamlines pass through the probe or around the exterior, with direct consequences for the collection of cloud water. By design, the streamlines exhibit the divergent pattern of a sub-isokinetic inlet, which tends to enhance the transmission of large cloud droplets relative to small aerosol particles. The critical design parameter is the mean

velocity ratio (to freestream) at the inlet, since this governs the flow rate in the probe and the extent of the enhancement of large droplets. The ratio of inlet velocity to freestream is affected by the effective total loss of the stator and other internal geometry (Table 2). Sensitivity simulations run with off-design conditions illustrate that a large fraction of the loss arises from swirl generation as well as the discharge from the pipe and points towards the potential for further design improvements, such as a divergent nozzle and/or a second stator as a flow straightener.

Extraction of water from the walls of the axial cyclone is challenging (SC04) and so a substantial focus of the design was the details of the capture zone. Capture zone simulations focus on the geometric details of the probe interior behind the stator at high spatial resolution. Regions of flow separation along the outer walls can result in the re-entrainment and subsequent loss of cloud water and should be avoided. Simulations are run with the pipe at the default (P10, furthest aft) position, as well as

two intermediate forward positions (P8 and P6). Streamline contours ahead and in the vicinity of the extraction slot for P10 (Fig. 4) do not show any regions of flow reversal along the outer wall, a feature usually indicative of flow separation, despite the adverse pressure gradient caused by the relatively stagnant conditions simulated in the extraction slot. A small separation bubble (green contour, Fig. 4b) is predicted within the extraction slot, which may cause some trapping and possible re-entrainment of water; however, the net flow is still directed towards the extraction port, along the outer periphery (red contour,

Fig. 4b). Any large accumulations of water trapped at the collection slot should be mitigated by the vacuum flow. Similar results are found for P8 and P6 (not shown), except that the separation bubble appears to become larger and the flow within the cavity becomes even less organized. This may be a result of the larger cavities and reduced penetration of the swirling flow and it appears that P10 is a more effective geometry, irrespective of other factors upstream.

### 3.2 Droplet Modelling

Droplet motions were simulated offline from the CFD using a Lagrangian trajectory model in MATLAB. In the model, droplet velocities are adjusted to flow field accelerations through a Stokes relaxation timescale, τ (see Eq. 7). The trajectories are therefore self-similar under conditions of constant Stokes Number, $Stk = \tau U l^{-1}$, where $l$ represents a geometric length scale.





Since $\tau \sim D^2$, droplet motions for different airspeeds can be scaled to the 115 ms$^{-1}$ nominal C-130 airspeed by deriving a C-130-equivalent droplet diameter, D$^*_{C130}$:

$$D^*{}_{C130} = D \left(\frac{U}{115}\right)^{1/2} \qquad (15)$$

In addition, a Markov Chain Monte Carlo prediction of the diffusional effects of turbulence on the droplet trajectory has been
included, using the CFD-predicted local turbulence kinetic energy (TKE). Random isotropic turbulent velocity components are generated with local temporal correlation determined by the turbulent length scale (fixed at 0.02 m) and local mean velocity. The fluctuations maintained the TKE from the CFD but did not attempt to model higher moments and would not have the correct inertial eddy spectrum; however, it is assumed that this would not affect the interpretation of the result. The effects of shatter and bounce were analyzed in idealized monodisperse laminar simulations. It was found that droplets bouncing off the
walls of the capture zone were not subsequently lost, even under elastic assumptions that could only be reached with extremely hydrophobic surfaces (Quéré, 2005). Including bounce on droplets impacting the stator surfaces could enhance the transmission of droplets through the stator, but with reduced axial momentum. The consequences of droplet shatter are difficult to predict because of the complexity of all the interior surfaces. However, we expect that similar shatter effects will be observed under conditions of constant Weber Number, $We = U^2 D \sigma^{-1}$, where σ is the droplet surface tension. The difference in $We$
and $Stk$ velocity scaling has implications for making droplet size comparisons between the C-130 and the Twin Otter.

Monodisperse simulations of the capture zone were used for design evaluation under two initial conditions. The first (Fig. 5a) represents droplets passing through the stator at equilibrium with the flow and then adjusting to the newly-formed swirl flow (equilibrium scenario). The second (Fig. 5b) illustrates a scenario where droplets have interacted with the stator, and we
simulate a limiting worst case from the point of view of potential capture, where droplets are released with zero momentum at the root of the blade trailing edge (stagnant scenario). It can be readily observed that, in the absence of turbulence, cloud droplets at typical diameters (20 µm) are captured by the outer walls, while 1-5 µm particles are centrifugally less affected by the cyclone in both scenarios. For even larger droplets, the equilibrium scenario results in trajectories which take too long to relax to the swirl and are not readily captured; however, in the stagnant scenario, large droplet trajectories asymptote to a limit
that is very close to the path represented by the 20 µm curve in Fig. 5b. However, large droplets are more likely to follow the stagnant scenario, because their inability to react to the swirl likely indicates their inability to navigate past the stator without impaction.

The addition of turbulence under the equilibrium scenario (Fig. 5c) illustrates a broadening of the possible trajectories at all
sizes. The end point for the ensemble of trajectories provides a statistical representation of the capture of droplets by the wall, and confirms the same pattern as Fig. 5a, although it is clear that 20 µm droplets are less affected by turbulence compared to 1-5 µm particles. Overall, the inclusion of turbulence predicts ~10% deposition of 1 µm aerosol particles on the capture zone walls, which would otherwise not be predicted by the laminar trajectories (Fig. 5a).





Extending the analysis of turbulent ensemble trajectories across all initial radial positions and all (C-130-equivalent) droplet diameters provides a statistical estimate of the size-dependent fraction of droplets that pass into the probe, traverse the stator, and reach the outer walls for collection (Fig. 6). Transmission efficiencies (Fig. 6a) for the inlet and stator reflect the fraction of cloud water that passes through each stage. In the case of the inlet, the lower asymptote reflects that of the inlet velocity ratio (Table 2), rising to complete transmission for very large droplets (e.g. precipitation). The stator transmission reflects the fraction of droplets that pass the stator without impaction with the remainder contacting the surface either on the blades or hub. Figure 6a implies that both scenarios can be expected for most cloud droplet sizes. For those droplets that impact, they may bounce off, shatter on or adhere to the surface. Although these processes do not necessitate a loss of cloud water, in the case of shatter or surface adhesion the microphysical identities of the droplets are lost, adding to the complexity of understanding the performance.

Downstream of the stator, we return to the equilibrium and stagnant scenarios (Fig. 6b and 6c, respectively) and illustrate the effect of the three pipe positions: P6, P8 and P10. The two scenarios reflect limiting conditions but the equilibrium scenario is close to what we would expect for droplets that are not impacted on the stator. Early in the design, the adjustable pipe was hypothesized to offer a means to control the "cut-size" of the cyclone; however, these results indicate that there is no obvious benefit for selecting P6 or P8 instead of P10. For the equilibrium scenario it only degrades performance and for the stagnant scenario the effect is modest and unlikely to be relevant because this scenario is more applicable to droplet impaction cases where the upstream microphysical information is lost. Figures 6b and 6c include the effects of turbulence; however, the capture of droplets is only degraded by < 5% compared to the equivalent laminar trajectory simulation (not shown), suggesting that turbulence in the probe interior is not hugely detrimental to its functionality.

A final aspect of the droplet and particle trajectory analysis is the potential for increased collisions amongst different sizes due to inertial separation of cloud droplets from the air streamlines. Collision and coalescence processes between cloud droplets are unlikely to alter the wall capture fractions shown in Fig. 6b and 6c since the combined water mass should still migrate to the walls under most scenarios. However, if droplets were to collect (scavenge) interstitial aerosols, it would potentially bias the composition of the cloud water samples. Using the conservative assumption that the volume swept by droplets on their path to the outer wall with respect to the total volume, which we assume supports the paths of interstitial particles, we can provide an upper bound estimate of the scavenging of interstitial particles within the confines of the probe. This volume fraction scales with LWC and inversely with the droplet diameter because it is a cross sectional effect. Even under "worst case" conditions of $1\text{g m}^{-3}$ LWC and 8 μm droplets, which could be possible under extreme aerosol loading such as pyrocumulus (Andreae et al., 2004), the estimated swept volume of the captured droplets is less than 0.8%. Thus, we can negate any contribution to the collected cloud water composition arising from interstitial aerosol.



### 3.3 Surface Processes

Cloud droplets, which impact and remain on the outer wall of the probe, need to migrate rearwards to the sample port before they are fully collected. Likewise, droplets that impact and adhere to the stator need to be transported towards the trailing edge before they are reintroduced into the airflow. The transport of water along the surface is governed by the near-surface air flow

and the interfacial stresses at the wall boundary, both of which affect, and are affected by, the geometry of the surface water. The dynamics of thin multi-phase fluid flows along a boundary are often described using lubrication theory applied to gravitational and shear driven cases (Hartley and Murgatroyd, 1964, Durbin, 1988, Moyle et al., 1999, Marshall and Ettema, 2005, Ding and Spelt, 2008, Fan et al., 2011). Three canonical regimes emerge: surface films, rivulets and isolated drops, with the expectation that the real flow may combine some of the characteristics of each.

Following extensive operation during a flight campaign, the surface of the capture zone was tested in the laboratory using the dynamic sessile drop technique to determine the water dynamic contact angle. The advancing and receding contact angles were found to be 100° and 80°, representing a moderately hydrophobic surface but approximately 20° less than was anticipated based on manufacturer data for the surface coatings. One possible reason is that the surface coating is not sufficiently resistant

to exposure to extended flight conditions and was partially eroded. A typical contact angle for untreated bare aluminum was assumed to be approximately 60°, with 20° hysteresis between advancing and receding angles (Hong et al, 1994, Samyn, 2014), but these angles are expected to be strongly affected by the amount of surface oxidation and the presence of contaminants (Trevoy and Johnson, 1958, Bernardin et al., 1997, Schuster et al, 2015).

The film regime assumes that the surface is completely wetted and water transport is driven by viscous shear. The bulk velocity of the film scales with the film thickness. The film mode is not stable below a critical thickness (Hartley and Murgatroyd, 1964) because under these conditions, dry patches will not become wetted and the flow is more likely to take on a rivulet structure. We document critical film thicknesses and associated bulk velocities in Table 3 for coated and uncoated surfaces and for typical C-130 and Twin Otter flight conditions. In all flight conditions, the wetting rate to maintain a stable film far

exceeds the possible supply of liquid water by more than an order of magnitude. This means that if a film were to form, it would be transient and highly unstable, thus the more plausible scenario is a partially wetted surface.

Over this range of contact angles, the dynamics of isolated droplet motion is expected to follow the so-called "creeping" regime instead of a low resistance "rolling" regime, which is possible for higher contact angles observed with super-hydrophobic

surfaces (Quéré, 2005). Future investigation into other available coatings may facilitate the rolling regime, with substantial reductions in interfacial forces; however, the benefits of any change would have to be considered in tandem with the durability, preservation, and potential artefacts resulting from such a surface finish exposed to flight conditions. For isolated droplets, static load analysis provides estimates of the critical drop effective diameter, beyond which interfacial forces caused by contact



angle hysteresis cannot resist the aerodynamic drag (Table 3). Also reported is the expected drop velocity in the creeping regime at the critical drop effective diameter. The drop velocities are considerably higher than expected for films of comparable thickness and can be attributed to the dominance of form drag affecting droplets compared to frictional shear in the case of the film. As the concentration of surface drops increases, the aerodynamic assumptions of an isolated drop are less

valid, since there is a shielding effect caused by surrounding drops and drop velocities will be lower. Based on these calculations, we expect cyclic behavior to emerge whereby dominant droplet clusters reach a critical size and are quickly swept along the surface capturing smaller downstream droplets in their path. The dry region in the wake of the departing drop or transient rivulet then starts accumulating small stable droplets which increase in density until droplet clusters start to form, resetting the cycle.

Despite the complexity of the details of the surface flow, we expect it to quickly relax to a steady state, where the axial gradient in wetting rate is balanced by the source of liquid water impacting the walls. This is assumed to occur once a critical volume of water is present on the walls (Table 3), which we estimate by assuming that sufficient critically sized droplets are present to satisfy the required wetting rate. During removal, they may appear temporarily as a rivulet, but it is not expected that these

would exist as a stable feature because they cannot be replenished fast enough. For simplicity, it is assumed that the wake of departing critically-sized drops is dry and the remaining area is covered with smaller quasi-stable drops distributed uniformly (by area) across all sizes down to typical cloud droplets. Further increases in wall volume in the form of a greater incidence of critically sized drops are buffered by their rapid removal. The critical wall volume represents an estimate of the residual water left uncollected after leaving cloud and constitutes a potential artefact for subsequent samples. We project that under

normal cloud conditions, it takes several seconds to accumulate these volumes which likely places a limit on reliably collecting cloud water in small (< 500 m horizontally) cumuliform clouds.

### 3.4 Evaporation

The flow through the probe is sub-isokinetic and therefore as the air is decelerated, droplets experience an increase in ambient pressure, particularly in regions where the flow is close to stagnation. These regions of compression are associated with a rise

in temperature, which reduces the relative humidity and promotes droplet evaporation. An opportune side effect of the compression heating is that the probe interior remains above the freezing point in slightly supercooled conditions, which extends the range of operation down to -5°C, for typical C-130 flight conditions. Unlike other mechanisms that result in under-sampling of the available cloud water, evaporation increases the concentration of non-volatile species and potentially liberates dissolved gases, making it difficult to conduct budgetary analysis of cloud water chemistry. SC04 provided calculations related

to evaporation and found that it was minor for their intended application. Here we revisit the issue for the current design and extend it to cover the full envelope of expected flight conditions. Much like SC04, in determining the total evaporation loss of the system, we consider two separate mechanisms. The first relates to the reduction in diameter of droplets as they traverse the sub-saturated regions within the probe caused by compression heating, and the second is the evaporation of water from the



walls of the capture zone exposed to the overlying stream of sub-saturated air. These processes are quantified in forthcoming efficiency calculations as a transmission efficiency against droplet evaporative loss, $\eta_{evap}$ and a wall water loss rate, $\delta_{evap}$. See Appendix A for full details of the derived analytical calculations used to estimate evaporative losses.

In these calculations, we conservatively assume that the freestream is isentropically and adiabatically brought to rest, yielding the maximum possible temperature increase in the absence of external sources, and hence these results can be interpreted as an upper bound to the evaporative losses. At the operating airspeeds and cloud conditions under consideration, the compression heating is sufficient to evaporate all the cloud water, if left to equilibrate. However, the droplet residence times are short enough such that droplet evaporation has a negligible influence on the background water vapor mixing ratio, which simplifies
the analysis.

The value of $\eta_{evap}$ is dependent on the ambient pressure, temperature, true airspeed, and droplet diameter, while $\delta_{evap}$ is independent of the microphysics. Increased operating temperature results in higher evaporation (Fig. 7) and is mainly driven by the Clausius-Clapeyron relationship controlling the vapor pressure-temperature slope. The primary effect of temperature is
to change the magnitude of vapor pressure gradients observed in the probe interior and so its effects are largely uncoupled from microphysical factors. Cloud droplets at 20°C, which represents an expected upper limit, are subjected to droplet evaporation losses $(1-\eta_{evap})$ approximately 1.8 times higher than at the minimum operating temperature (-5°C), while wall losses also increase with ambient temperature (Fig. 7b). The effects of pressure are negligible (~1% change) when treated in isolation, but operating at higher altitudes reduces the range of possible temperatures, which is the more dominant effect.
Increasing the airspeed results in more evaporation because the compression heating is greater. Linear perturbation analysis derived in Appendix A suggests approximately linear scaling of droplet evaporative losses with airspeed, when the Mach Number is less than 0.35. In contrast, the wall loss rate is more sensitive to airspeed with cubic dependence. These approximate rules can be used to scale Fig. 7 for aircraft operating at lower airspeed.

Droplet size only affects the droplet evaporation fraction with smaller droplets more readily evaporated before capture by the walls (Fig. 7a). In contrast, wall loss rates are not microphysically dependent; however, if normalized by the fraction sampled (equivalent to an efficiency) there is an inverse dependence with LWC, such that wall losses are more relevant to low LWC cases. An illustration of this is shown in Fig. 7b for LWC = 0.2 g m$^{-3}$. In deeper clouds (generally higher LWC), the combined effects of evaporation are not substantial enough to add uncertainty to the analysis of absolute species concentrations.
However, for samples collected in low LWC environments and at small droplet sizes, such as typically found at cloud base, evaporation should be accounted for in deriving absolute concentrations and has particular relevance for assessing the vertical structure of trace species in cloud. This analysis also suggests that cloud water collection becomes substantially more difficult when LWC < 0.1 gm$^{-3}$ because of the compounding effect that evaporation has on the length of time it takes to collect a sample.





### 3.5 Collection Efficiency

We define total collection efficiency (TCE) as the ratio of observed (or model predicted) cloud water collection rate (CR) to the freestream cloud water flux across an area equivalent to the inlet cross section ($A_{inlet} = 3.85 \times 10^{-3}\,\mathrm{m^2}$) and at freestream true air speed, $U_\infty$, equivalent to the definition of "collectable water" given in characterizing the slotted-rod performance (Kim and Boatman, 1992):

$$TCE = \frac{\rho_w \cdot CR}{LWC \cdot U_\infty \cdot A_{inlet}} \tag{16}$$

The CR measured during flight can be calculated for each sample from the total volume collected and the time in cloud (in which case the terms above are replaced with sample time-average quantities). Near real-time CR can be derived from the water metering system, allowing TCE to be measured at higher temporal resolution. Alternatively, CR can be estimated by combining the results of the particle tracking, evaporation and surface transport models. Size-dependent transmission efficiencies for the inlet, $\eta_i$, and stator, $\eta_s$, are combined with the wall capture fraction, $\eta_c$, droplet evaporation, $\eta_{evap}$, and wall evaporation loss, $\delta_{evap}$. For the fraction that passes the stator without impaction ($\eta_s$) we use the equilibrium scenario (Fig. 6b) to represent $\eta_c$. Since it is not possible to explicitly constrain the downstream sizes of stator-impacted droplets, we empirically parameterize this with a non-size-dependent coefficient, C:

$$CR_{model} = \left[ U_\infty A_{inlet} \int \eta_i \eta_{evap} (\eta_s \eta_c + (1 - \eta_s) C) \mathcal{V}(D) dD \right] - \delta_{evap} \tag{17}$$

where $\mathcal{V}(D)$ is the ambient volume size distribution of droplets satisfying:

$$\rho_w \int \mathcal{V}(D) dD = LWC \tag{18}$$

The TCE can now be predicted if a drop size distribution is assumed. For convenience in comparing the simulations with observations, we construct a volume size distribution from an effective diameter, LWC, and a dimensionless effective variance of 0.06, which was found to be typical of the observed cloud droplet size distributions. With more flight data across a larger dynamic range, we may be able to invert the TCE relationship to retrieve size dependent efficiency parameters; however, at present we are limited to comparing the measured TCE with the current model of TCE. The "C" coefficient incorporates the capture of droplets that have impacted the stator, accumulated and then been swept to the trailing edge where they migrate to the walls in accordance with Fig. 6c. The accumulated water on the stator would likely be subjected to similar interfacial and aerodynamic forces as the capture zone walls, and so by inspection of Table 3, these shed drops are expected to be large and fall on the right asymptote of Fig. 6c, implying values in the range 0.9 to 1.0. However, C also includes the fate of shattered droplets that may include fragments that are not subsequently accumulated on surfaces. This is expected to result in some loss





for small fragments < 5 µm, although observations of shatter (Quéré, 2005) generally do not indicate complete fragmentation of drops; rather, the main drop remains with some mass expelled into a cluster of small satellite droplets.

## 4 Flight Performance

The AC3 was integrated onto the NASA C-130 for the second deployment of the North Atlantic Aerosol and Marine Ecosystems Study (NAAMES) in May 2016. The probe was mounted on a standoff pylon approximately 12 inches from the fuselage skin next to aerosol and gas inlets and forward of the propeller. Nine research flights (RF) were conducted in the marine boundary layer over the central North Atlantic, resulting in 62 cloud water samples. During July 2016, the AC3 was integrated on the Center for Interdisciplinary Remotely-Piloted Aircraft Studies (CIRPAS) Twin Otter based in Marina, California for the Fog and Stratocumulus Evolution (FASE) campaign (Sorooshian et al., 2018). The probe was mounted underneath the inboard wing pylon. A total of 156 cloud water samples were collected during fifteen research flights. The AC3 was then reintegrated on the NASA C-130 in a similar configuration for the third NAAMES deployment in September 2017. The operable shutter system was added to the probe configuration for this deployment and it was at this point that the probe was treated with the Teflon hardcoat and fluoropolymer surface coating. The flow metering and auto-collection was also added for this campaign. During the third NAAMES deployment, eight research flights provided 57 samples.

Table 4 provides a parameter summary for each research flight. Environmental conditions (e.g. temperature) were calculated using a LWC-weighted mean over the duration of the sample and the ranges refer to the highest and lowest sample per flight. All cloud water samples were from marine warm clouds (i.e. comprising of liquid water) and spanned a temperature range of -9°C to 12°C and an altitude range of 100 m to 3600 m. Sampling was attempted at ambient temperatures lower than -9°C, but there was no collection. Between -9°C and -5°C collection was very intermittent and not dependable and sometimes not correlated with cloud penetration; it was concluded that in these conditions, supercooled water froze onto the interior of the probe or collection system. The minimum operating temperature therefore agrees closely with the -5°C lower cut off derived from compression heating calculations (see Sect. 3.4). Histograms of cloud variables and CR (Fig. 8) highlight the range of cloud conditions seen within each campaign and the associated range of expected collector performance. Data for NAAMES-3 conveys the performance for the most refined state of the system where the median CR was 4.5 mL min$^{-1}$ for 0.22 g m$^{-3}$ LWC and 25 µm effective diameter droplets (both median values). At this LWC, the collectable water was 5.8 mL min$^{-1}$, implying a TCE of 77%.

Statistics of TCE are calculated for discrete bins of droplet effective diameter for each campaign (Fig. 9). The FASE data are stratified according to the pipe position. The trends in the TCE demonstrate an acceptable match to the predicted behavior; however, the level of variability indicates that there are parts of the process that are still not fully understood. Focusing first on the NAAMES-3 data: the assumption that the "C" coefficient be near unity appears to be appropriate, although there is





increasing discrepancy for smaller diameters. The use of Stokes Number scaling (Eq. 15) seems to be appropriate in interpreting the P10 Twin Otter results, in that it produces consistent model agreement, with respect to the choice of C. Furthermore, if C had a Weber number dependence, this result would suggest that it ought to be quite weak. Flight data also confirms the modelling conclusion that there is little value in operating at P8 or P6 compared to P10. Although not shown in
Fig. 9, the flight data TCE for P8 and P6 are lower than the model predictions, suggesting additional issues with having the pipe further forward not captured by the model.

A contributing source of uncertainty that may play into the variability of individual TCE estimates is the LWC, which was derived from the droplet size distribution measured by the Cloud Droplet Probe (Droplet Measurement Technologies) and does
not include contributions from droplets exceeding 50 μm. Consequently, TCE in precipitating clouds may be over-predicted and likely explains data points which exceed TCE = 1. An uncertainty for NAAMES-2 and FASE was assessing the appropriate sample start and stop times for broken cloud scenes; however, this was improved for NAAMES-3 with the inclusion of the water metering system.

The water metering system was particularly useful for quantifying instantaneous CR during a cold air outbreak event, which was flown on RF6, RF7 and RF8 during NAAMES-3. The flight tracks and location of cloud samples (Fig. 10) spanned a large region (> 1000 km) of the North Atlantic and captured a range of different marine boundary layer cloud types. The three example cloud water samples shown in Fig. 11 are associated with the three respective cloud scenes shown in Fig. 10 and illustrate the agreement in the variability of LWC with the resulting CR. The time base for the water metering system has been
adjusted for the collection lag, which is approximately 10 s, and the CR data was smoothed using a 5 s moving average filter. These selected samples were collected in clouds that did not have substantial contributions from precipitation and showed strong positive correlation between LWC and CR ($R^2 > 0.7$). There was significant noise in the CR data at scales < 5 s, which we attribute to intermittency of collection and is in general agreement with the expectation of a critical volume of cloud water inside the probe (Table 3). This limit provides an estimated cutoff for the fastest time resolution that could be achieved by any
prospective online or offline chemical analysis technique.

## 5 Cloud chemistry intercomparison

During FASE, the Twin Otter, was also equipped with a slotted-rod collector and used to validate the samples collected by the AC3. The slotted-rod has been used on the Twin Otter for other studies during airborne field campaigns in the same region (Hegg et al, 2002, Sorooshian et al., 2013, Prabhakar et al., 2014, Wang et al., 2014, Youn et al., 2015, Sorooshian et al., 2015,
Crosbie et al, 2016). Samples were typically collected during level, in-cloud flight legs, with the two systems operating in parallel with coordinated sampling, when possible. The sampling duration varied depending on the cloud conditions and sampling techniques in order to obtain an excess of 10 mL per sample, where possible, allowing for multiple analyses. Samples



from both collectors were analyzed for ionic composition (Ion Chromatography, IC; Thermo Scientific Dionex ICS-2100 system) and elemental composition (triple quadrupole inductively coupled plasma mass spectrometry, ICP-MS; Agilent 8800 Series) at the University of Arizona. During the FASE campaign, there were a total of 156 slotted-rod samples to compare against the 157 from the AC3. Trace species concentrations in the cloud water were converted to estimated air-equivalent

mass concentrations using the mean LWC over the duration of the sample and are reported alongside aqueous mass concentrations (Table 5).

Campaign-mean cloud water constituents for the two systems (Table 5) show good agreement for the majority of the trace species. Species are sorted by abundance (quantified by the median aqueous mass concentration measured by the slotted-rod)

and comparisons are presented for species that exceed 15 μg L$^{-1}$. Chloride (Cl$^-$) and sodium (Na$^+$) ions are the two largest constituents that were measured in the cloud water and reflect the dominant contribution of sea salt to the cloud-activated aerosol mass in the marine boundary layer, consistent with other studies (Straub et al., 2007; Benedict et al. 2012; Wang et al. 2016). The concentration of major sea salt ions, including Cl$^-$, Na$^+$, sulfate (SO$_4^{2-}$), magnesium (Mg$^{2+}$), and calcium (Ca$^{2+}$), is elevated in the AC3 samples compared to the slotted-rod by a factor ranging between 1.2 and 1.8. However, the Cl$^-$:Na$^+$ and

Mg$^{2+}$:Na$^+$ mass ratios match reported data for sea water composition (Seinfeld and Pandis, 2016), with good agreement between the probes. A contributing factor for the offset in absolute concentration could stem from differences in the collection of small droplets between the two collectors, since species concentrations often exhibit dependence on droplet size (e.g. Bator and Collett, 1997). The 6061-T6 aluminum and 316 stainless steel used for the body of the probe could explain the relative enhancements in elemental aluminum (Al) and iron (Fe), as well as potential contributions to the Mg$^{2+}$ enhancement. The

Teflon anodized hardcoat, which was added after this intercomparison, should help mitigate any artefacts from the alloys. The Ca$^{2+}$:Na$^+$ ratio is enhanced when continental airmasses are mixed into the marine boundary layer, representing contributions from dust. The AC3 samples are more elevated above sea salt than the slotted-rod, but the overall agreement is still encouraging (<20% difference). The SO$_4^{2-}$:Cl$^-$ ratio is often used to identify non-sea salt sulfate (nss-SO$_4^{2-}$) sources and the sulfate enhancement above sea water is captured in close agreement by both probes. In this region, sulfate sources other than

sea salt may be associated with ship emissions, marine biota or continental sources (Sorooshian et al, 2009, Coggon et al, 2012, Coggon et al, 2014, Wang et al., 2014). Since methanesulfonate (MSA) is a product in the oxidation of biogenic sulfur emissions to SO$_4^{2-}$, the MSA:nss-SO$_4^{2-}$ ratio can be used to separate marine biogenic sources and this is also closely replicated between the probes.

Trends and variability in cloud water concentration within and between flights are examined (Fig. 12) by capturing the temporal overlap between the two sample sets. Since the collectors were not always synchronized in their operation or sample duration, we assign an index to all pairs of samples that overlap such that a sample may be assigned multiple indices if spanning multiple samples of the other probe. Samples which do not coincide in time with the other probe are omitted from this comparison.



For convenience, we name the resulting temporally coincident indices a "pseudo-time series", since it represents a chronology of all the concurrent samples across the campaign.

As an example, we show chloride, nitrate and calcium (Fig. 12 a-c), and three ion ratios: $Cl^-$:$Na^+$, $SO_4^{2-}$:$Cl^-$ and $Ca^{2+}$:$Na^+$ (Fig.

12 d-f, respectively). Despite a range spanning several orders-of-magnitude, the major variability in $Cl^-$, $NO_3^-$ and $Ca^{2+}$ aerosol-equivalent concentrations is satisfactorily replicated between the two techniques. Two notable exceptions are during RF4, where the AC3 $Cl^-$ is higher than the slotted-rod by a factor of six (median of all sample pairs) and during RF15 and RF16 where the AC3 $Ca^{2+}$ exceeded the slotted-rod by a factor of fourteen. These instances were likely caused by ground contamination of the AC3 probe, which we believe has been mitigated through subsequent improvements in the pre-flight

procedures. The intercomparison was also conducted prior to the installation of the operable shutter, which may explain some of the enhancements. Some flights, most notably RF6, RF9 and RF15, show large variability between consecutive samples and is generally captured by both probes in good agreement although, qualitatively, the oscillations seen in the AC3 pseudo-time series are less extreme. Broad trends across a flight (e.g. RF10) are well captured by both probes.

The $Cl^-$:$Na^+$ ratio agrees with the expected sea salt mass ratio (1.80) for most flights, but there are also times (e.g. RF15 and RF16) where significant reduction occurs and may be associated with chloride depletion mechanisms where acids drive chloride into the gas phase (McInnes et al, 1994, Zhuang et al, 1999). These broad patterns are closely replicated between the two probes. In addition, when sources other than sea salt become more influential (e.g. biomass burning), there is potential for peripheral enhancements to shift the ratio. A possible example of this is seen in RF4, where there is more variability

although the magnitudes of the perturbations vary between probes. Enhancements in $SO_4^{2-}$:$Cl^-$ above 0.14 are indicative of non-sea salt sulfate. There is good agreement, again with the exception of RF4, where the lower ratio measured by the AC3 is more attributed to an overestimate of chloride than an underestimate of sulfate. In the later flights, an increase in the abundance of non-sea salt sources is seen by both probes. Despite the substantial variability in chloride during RF6, RF9 and RF15 that was noted earlier and presented a challenge to correlating both probes, these flights show excellent agreement in the

$SO_4^{2-}$:$Cl^-$ ratio and may indicate that relative abundances can be accurately measured during sampling conditions that are challenging in achieving accurate absolute concentrations.

The nearby Soberanes Fire affected the region starting on RF4 and remained active for the rest of the campaign. The plume was transported offshore on a number of the flights and was likely to be a significant conduit for lofting near surface continental

airmasses into the lower free troposphere, where it could later descend into the marine cloud deck. It is also known that fires can locally drive substantial dust and soil emissions (Kavouras et al, 2012, Diapouli et al, 2014, Maudlin et al., 2015, Schlosser et al., 2017). Comparing the $Ca^{2+}$:$Na^+$ ratio between probes shows that RF4 exhibits a marked difference in the relative abundance of non-sea salt calcium compared to the other flights. The approximately hundred-fold increase seen during this flight was observed in samples from both probes. During the flights when chloride was high (RF3, 6-9, 12), there were



generally low $Ca^{2+}$:$Na^+$ ratios that approached the value for sea salt (0.04), indicating that terrestrial sources were insignificant. However, there was a noteworthy pattern that emerged at the beginning of several flights where the $Ca^{2+}$:$Na^+$ ratio started high and then decreased through the flight. In some cases (e.g. RF3), it occurred only for the AC3, while in other instances it occurred for both probes (e.g. RF5, RF6, RF11), with the more significant effect varying between probes. While it is possible

that some of the gradient could be explained by a diminishing influence of fire-augmented terrestrial sources with offshore distance, it is likely that part of the effect arises from contamination accumulated out of cloud. The material that adheres to the probe prior to cloud sampling then takes some time to be removed from interior surfaces.

Particles that activate as cloud condensation nuclei or that are scavenged within the cloud contribute to the bulk chemistry of

the cloud water and, therefore, their inclusion in the sample is desired. However, these particles can be sufficiently large to be collected by either cloud water system outside cloud or in low LWC regions. In particular, aerosol haze droplets encountered near cloud edges and below bases where the relative humidity is high but below saturation present a challenge. In these environments, the particles are large enough to be readily captured by the walls, but the relatively low amount of liquid water precludes reaching critical wall volumes needed for sample collection. Instead, large particles that are accumulated on the

collector can remain until a cloud is encountered at a later time, at which point the accumulated material may be reintroduced into a subsequent sample. In addition, upon leaving cloud, unsampled water that evaporates within the probe may leave insoluble particles and residual solute mass on the collector. These issues have a potential impact on both collectors, but the impact of clear air contamination is generally minimized with the slotted-rod by taking the probe out of the airstream while not in cloud. The operable shutter was added to the AC3 to avoid clear air contamination; however, this had not been included

at the time of the FASE campaign. We attribute the enhancement of sea salt and dust particle constituents seen in the AC3 samples (Table 5) to an increased exposure to clear air contamination compared to the slotted-rod. The improved AC3 design incorporating the shutter follows the strategy employed by SC04, and its implementation during NAAMES-3 indicates that this clear air contamination issue will not present a substantial ongoing limitation.

The performance of the shutter was tested during NAAMES-3 by comparing blank samples collected immediately after landing on RF6, RF7 and RF8 with the last in-flight sample. The interior of the probe was sprayed with ultrapure water to coat all surfaces and then the first 5 mL of the resulting sample was collected. Using $Na^+$ as a representative tracer, it was found that post-flight blank concentrations varied between 4-13% of the last in-flight sample. The majority of cloud water samples collected during NAAMES-2 were found to be unreliable for chemical speciation because a large fraction of the flight time

was spent in the marine boundary layer below cloud and, without the shutter, the effect of sea salt accumulation was so significant that the cloud water signal was lost within the high background. Successive pre-flight rinses with ultrapure water were performed during NAAMES-2 and we use a sample of the first of these rinses as a proxy for the post-flight blank of the prior flight. A comparison with the post-flight blanks collected during NAAMES-3 indicates that without the shutter, the residual contamination due to sea salt could be higher by a factor of 12.8±5.9, as quantified by $Na^+$ concentration.



## 6 Summary

Performance of the AC3 has been evaluated through modelling and flight testing. We demonstrate a reliable method of collecting cloud water samples for offline chemical analysis. CFD simulations coupled with particle tracking and multiphase surface transport modeling provide detailed estimates of the probe performance across the entire range of potential operating conditions. The collection efficiency is validated through the analysis of in-flight collection rates from 275 discrete samples spanning a wide range of cloud and flight conditions. We intend to extend this by continuing to include data from current and future field campaigns, particularly from polluted cumuliform clouds, which are underrepresented in the current archive. Samples collected during the FASE campaign were compared with samples collected using the slotted-rod collector. Comparison of trace species concentrations showed a good agreement between collection techniques. The addition of the operable shutter for NAAMES-3 significantly reduced the collection of aerosol particles and haze droplets during flight outside of clouds, which was confirmed by analyzing post-flight field blanks. The surface treatments applied to the probe before the NAAMES-3 campaign may also reduce the presence of surface contaminants and improve collection efficiency.

## Acknowledgements

This research was funded by NASA's Radiation Sciences and Tropospheric Chemistry Programs, as well as NASA's Earth Venture-2 Program through the Earth System Science Pathfinder (ESSP) Program Office. Twin Otter campaigns and data analysis were supported by Office of Naval Research grants N00014-10-1-0811, N00014-11-1-0783, N00014-10-1-0200, N00014-04-1-0118, and N00014-16-1-2567. MDB acknowledges support from the NASA Postdoctoral Program, ABM acknowledges support from the Mexican National Council for Science and Technology (CONACyT). We wish to thank the ESSP Program Office and M. Kleb for their support throughout the NAAMES deployments. The authors would also like to acknowledge the contributions of B. Beaton, T. Clark, R. E. Dyke, D. Fahringer, R. Wagner, and W. Welch during the probe design and fabrication. We also wish to thank M. Chance and M. Nowicki for their support related to C-130 engineering and operations, H. Jonsson for support with Twin Otter operations, and we would like to express deep appreciation to the pilots and flight crew of both aircraft. The authors also acknowledge the helpful review of the manuscript by M. Guzman.

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

**Appendix A**

The effects of evaporation are considered through two mechanisms: Part 1 considers the droplet during the time-of-flight from
entry into the probe and arrival at a collection surface, while Part 2 considers evaporation from water on the surface.

Part 1: Droplet Evaporation
Droplets in the airstream entering the probe are subjected to an increase in temperature and pressure as the flow is decelerated following an adiabatic and isentropic trajectory from initial state (subscript 0) to final state (subscript 1), which we assume is
at stagnation:

$$T_1 = T_0 \left( 1 + \frac{\gamma - 1}{2} M_0{}^2 \right) \tag{A1}$$





$$P_1 = P_0 \left(\frac{T_1}{T_0}\right)^{\frac{\gamma}{\gamma-1}} \tag{A2}$$

where T is the static temperature, P is the static pressure, and $\gamma$ is the ratio of specific heats of air. Mach Number is defined $M = U\sqrt{\gamma R_a T}$ where $R_a$ is the specific gas constant for air. While there is still a bulk velocity $U_1$ within the probe, the pressure and temperature rise are calculated at stagnation, since it is assumed that some regions within the interior will be close to

stagnation and these regions reflect a potential "worst case" for evaporative losses. The value of $U_1$ is estimated from CFD and/or the analytical model developed in Sect. 2.2, together with the upstream $U_0$.

The saturation ratio of the final state, S, can be determined from the temperature and pressure, using the assumption that the initial state was saturated:

$$S = \frac{e_1}{e_{1,s}} = \frac{e_s(T_0)}{e_s(T_1)}\left(\frac{P_1}{P_0}\right) \tag{A3}$$

The saturation vapor pressure, $e_s$, is determined numerically from the Clausius-Clapeyron relationship:

$$\frac{de_s}{dT} = \frac{l_v e_s}{R_v T^2} \tag{A4}$$

where $l_v$ is the latent heat of vaporization of water and $R_v$ is the specific gas constant for water vapor. The growth/shrinkage rate of a droplet, exposed to S, can be related back to the environmental variables that define the initial state (which we define as the state function $\xi$) using a modification of the expression derived in Seinfeld and Pandis (2016):

$$\frac{D_p}{4}\frac{dD_p}{dt} = \frac{S-1}{F_k + F_d} = \xi(P_0, T_0, M_0) \tag{A5}$$

where drop diameter is denoted $D_p$, and the functions $F_k$ and $F_d$ represent diffusional fluxes related to the effects of heat and water vapor mass, respectively:

$$F_k = \left(\frac{l_v}{R_v T_1} - 1\right)\frac{\rho_w l_v}{k T_1} \tag{A6}$$

$$F_d = \frac{\rho_w R_v T_1}{D_v e_s(T_1)} \tag{A7}$$

where $\rho_w = 1000$ kg m$^{-3}$ is the density of water, $D_v$ is the diffusivity of vapor into air, and k is the thermal conductivity of air, which are expressed as:

$$D_v = \frac{2.11}{P_1}\left(\frac{T_1}{273}\right)^{1.94} \tag{A8}$$

$$k = 4.39 \times 10^{-3} + 7.1 \times 10^{-5} T_1 \tag{A9}$$

By assuming that a droplet is exposed to the final state for a characteristic time scale $\tau = L/U_1$, this allows us to estimate of the final droplet diameter:

$$D_{p,1} = \left(D_{p,0}^2 + 8\xi\tau\right)^{\frac{1}{2}} \tag{A10}$$





Defining the efficiency of capture with respect to droplet evaporation as the ratio of final liquid water content (LWC) to initial LWC:

$$\eta_{evap} = \frac{LWC_1}{LWC_0} = \frac{D_{p,1}{}^3}{D_{p,0}{}^3} = \frac{(D_{p,0}{}^2 + 8\xi\tau)^{\frac{3}{2}}}{D_{p,0}{}^3}$$

(A11)

Part 2: Wall Evaporation

Evaporation from wall surfaces is governed by convective mass transfer across the surface boundary layer. The growth and structure of the surface boundary layer is determined by the local Reynolds number, $Re_x = Ux/\nu$, where $\nu$ is the momentum diffusivity. For the upstream section, axial flow prevails and so the region from the inlet to the stator trailing edge is modelled

using the inlet velocity, $U_1$, and axial length of this region as U and x, respectively. For the swirl region, we modify the length and velocity scales to accommodate the local swirl streamlines. For $M_0$=0.36, $Re_x$ values at the trailing edge and the end of the capture zone are estimated at 3.1E+05 and 5.2E+05, respectively. We use a midpoint $Re_x$ = 4.1E+05 in this analysis. The velocity coefficient associated with convective mass transfer of water vapor in the turbulent boundary layer can be approximated for a flat plate according to Kays and Crawford (1993):

$$c_{H_2O} = 0.0287 Re_x{}^{-0.2} Sc^{-0.4}$$

(A12)

where $Sc = \nu/D_v$ is the Schmidt number for water vapor at these conditions. Although the $Re_x$ suggests that the flow is close to the critical value for laminar-to-turbulent transition, we use the turbulent expression for estimating $c_{H2O}$ since it is the worst case and in keeping with estimating an upper bound on evaporative losses. We also assume that the entire capture zone contains surface drops with surface area $A_{wall}$. The mass loss rate due to wall evaporation can be written:

$$\delta_{evap} = c_{H_2O} U_1 A_{wall} (1 - S) \frac{e_s(T_1)}{R_v T_1}$$

(A13)

Linearized analysis and scaling

Linearizing Eq. A1-A3 by assuming the perturbation between initial and final state is small combined with Eq. A4, gives:

$$S \approx 1 - \frac{\Delta e_s}{e_s} + \frac{\Delta P}{P} \approx 1 - \frac{l_v \Delta T}{R_v T^2} + \frac{\gamma \Delta T}{(\gamma - 1)T}$$

(A14)

and is suitable when M<0.35 but becomes increasingly more accurate as $M \to 0$. Considering Eq. A5, and using $\Delta T \sim M^2$ from Eq. A1 gives $\xi \sim f(P,T)M^2$ where the dependence on T involves a number of competing terms from Eq. A6, A7 and A14 and the dependence on P arises (weakly) from an inverse scaling with vapor diffusivity $D_v$. Considering 268K to 293K as the temperature range and 1000 hPa to 500 hPa as the pressure range we can approximate the dependence $f \sim -P^{-0.34} T^{6.18}$.

Assuming evaporative losses amount to a small fraction of the volume, Eq A11 can be linearized to:

$$1 - \eta_{evap} \approx -\frac{12\xi\tau}{D_p{}^2}$$

(A15)





using $\tau \sim U^{-1}$ and $M \sim U$ we arrive at the approximate relation:

$$1 - \eta_{evap} \sim T^{6.18} P^{-0.34} D_p^{-2} U \qquad (A16)$$

5    For wall evaporation, we consider the terms in Eq. A13, ignoring the quite weak dependence of $c_{H2O}$ and the weak dependence of $(1-S)$ on T. The dependence of $e_s$ on T is highly non-linear and so we estimate it empirically over the above range, yielding $e_s \sim T^{19.2}$. Substituting into Eq A13:

$$\delta_{evap} \sim U^3 T^{18.2} \qquad (A17)$$



Table 1: Summary of probe specifications

| Category | Property | Units | Value |
|---|---|---|---|
| Bulk properties | Mass | kg | 4.5 |
| | Dimensions: *l x w x h* | m | 0.5 x 0.15 x 0.2 |
| Connections | Plumbing | | Sample: 1/8"PFA with Swagelok® bulkhead connection, Vacuum: 3/8" polyethylene with barb connection |
| | Electrical | | 11-pin 12V,5V power and data |
| Specific dimensions (see Fig. 1) | L | mm | 12 – 76 |
| | $L_{INLET}$ | | 100 |
| | $L_{BLADE}$ | | 50 |
| | R | | 40.5 |
| | $R_{INLET}$ | | 35 |
| | $R_{PIPE}$ | | 27.5 |
| Structural | TAS | m s$^{-1}$ | <250 |
| | OAT | °C | -40/+40 |

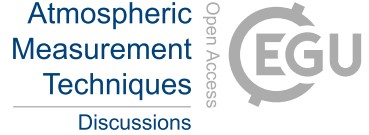

Table 2: Relationship between inlet velocity ratio and interior losses

| Case | $K_{eff}$ | $U_{INLET}/U_\infty$ |
|------|-----------|----------------------|
| No stator | 0.03 | 0.537 |
| No swirl | 0.53 | 0.497 |
| **Design** | **2.14** | **0.411** |
| High loss | 3.5 | 0.364 |





Table 3: Summary of critical interior surface conditions

|  |  | C-130 | | Twin Otter | |
|---|---|---|---|---|---|
|  |  | Uncoated | Coated | Uncoated | Coated |
| TAS | $(ms^{-1})$ | 115 | | 50 | |
| $\theta$ | (°) | 60 | 90 | 60 | 90 |
| $\Delta\theta$ | (°) | 20 | 20 | 20 | 20 |
| $h_{film}$ | (μm) | 245 | 297 | 788 | 956 |
| $V_{film}$ | $(ms^{-1})$ | 0.54 | 0.65 | 0.30 | 0.36 |
| $d_{eff}$ | (μm) | 191 | 148 | 768 | 627 |
| $L_{drop}$ | (μm) | 307 | 186 | 1235 | 790 |
| $H_{drop}$ | (μm) | 89 | 93 | 356 | 395 |
| $V_{drop}$ | $(ms^{-1})$ | 4.3 | 6.5 | 3.1 | 4.3 |
| Vol. | (μl) | 495 | 625 | 1991 | 2647 |

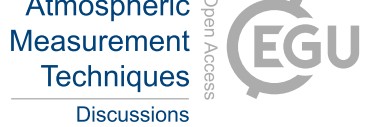

Table 4: Summary of flights and cloud water samples

|  | Flight | Date | Sample Count | Total Volume | Temperature Min/Max | Altitude Min/Max |
|---|---|---|---|---|---|---|
|  |  |  |  | (ml) | (°C) | (m) |
| NAAMES-2 | 1 | 05/18 | 8 | 36 | -7.1 / -0.4 | 540 / 2150 |
| (NASA C-130) | 2 | 05/19 | 9 | 30 | -2.3 / 2.3 | 270 / 970 |
|  | 3 | 05/20 | 11 | 58 | -0.8 / 0.8 | 440 / 690 |
| 2016 | 5 | 05/27 | 3 | 6 | -9.4 / -7.5 | 1230 / 1460 |
|  | 6 | 05/28 | 14 | 66 | -5.1 / -0.6 | 600 / 1290 |
|  | 8 | 05/30 | 13 | 55 | -2.9 / -0.3 | 1200 / 1550 |
|  | 9 | 06/01 | 2 | 13 | -4.7 / -2.8 | 3040 / 3610 |
| FASE | 2 | 07/22 | 16 | 244 | 10.4 / 11.7 | 100 / 280 |
| (CIRPAS Twin Otter) | 3 | 07/25 | 16 | 56 | 10.2 / 12.1 | 190 / 620 |
|  | 4 | 07/26 | 10 | 56 | 9.5 / 10.5 | 100 / 130 |
| 2016 | 5 | 07/27 | 12 | 89 | 9.6 / 11.2 | 160 / 280 |
|  | 6 | 07/29 | 29 | 120 | 9.8 / 12.7 | 120 / 480 |
|  | 7 | 08/01 | 4 | 52 | 9.6 / 10.1 | 380 / 400 |
|  | 8 | 08/02 | 4 | 19 | 10.3 / 11.5 | 310 / 390 |
|  | 9 | 08/03 | 4 | 39 | 10.8 / 11.1 | 350 / 380 |
|  | 10 | 08/04 | 13 | 113 | 10.8 / 11.4 | 400 / 530 |
|  | 11 | 08/05 | 15 | 163 | 8.6 / 10.0 | 430 / 750 |
|  | 12 | 08/08 | 1 | 2 | 12.3 | 380 |
|  | 13 | 08/09 | 12 | 66 | 9.9 / 10.9 | 300 / 640 |
|  | 14 | 08/10 | 6 | 53 | 9.5 / 10.1 | 370 / 490 |
|  | 15 | 08/11 | 9 | 83 | 9.8 / 11.5 | 200 / 460 |
|  | 16 | 08/12 | 6 | 27 | 10.9 / 11.8 | 280 / 410 |
| NAAMES-3 | 2 | 09/06 | 4 | 6 | 14.6 / 16.7 | 150 / 360 |
| (NASA C-130) | 3 | 09/08 | 3 | 12 | 19.0 / 19.5 | 250 / 750 |
|  | 4 | 09/09 | 5 | 33 | 11.5 / 18.9 | 150 / 1600 |
| 2017 | 5 | 09/12 | 1 | 1 | 4.0 | 850 |
|  | 6 | 09/16 | 15 | 63 | -2.5 / 8.0 | 660 / 1960 |
|  | 7 | 09/17 | 14 | 51 | -4.1 / 4.9 | 780 / 1540 |
|  | 8 | 09/19 | 12 | 67 | -2.1 / 3.3 | 1240 / 1970 |



Table 5: FASE campaign major species mass concentrations and selected ratios between the AC3 and the slotted rod (SR)

| Species | Median aqueous mass concentration | | | Median aerosol equivalent mass concentration | | |
|---|---|---|---|---|---|---|
| | SR | AC3 | AC3/SR | SR | AC3 | AC3/SR |
| | $\mu g\ L^{-1}$ | | | $ng\ m^{-3}$ | | |
| $Cl^-$ | 5293 | 6641 | 1.25 | 1578 | 2074 | 1.31 |
| $Na^+$ | 2989 | 3789 | 1.27 | 910 | 1157 | 1.27 |
| $SO_4^{2-}$ | 2153 | 2748 | 1.28 | 549 | 842 | 1.53 |
| $NO_3^-$ | 1011 | 1276 | 1.26 | 305 | 375 | 1.23 |
| $Mg^{2+}$ | 372 | 694 | 1.87 | 115 | 216 | 1.88 |
| MSA | 322 | 394 | 1.23 | 89 | 116 | 1.30 |
| $Ca^{2+}$ | 267 | 437 | 1.64 | 81 | 131 | 1.62 |
| oxalate | 233 | 279 | 1.20 | 61 | 76 | 1.24 |
| $NH_4^+$ | 200 | 215 | 1.07 | 51 | 69 | 1.35 |
| malonate | 126 | 123 | 0.97 | 31 | 35 | 1.16 |
| $K^+$ | 105 | 164 | 1.57 | 31 | 53 | 1.69 |
| pyruvate | 84 | 131 | 1.57 | 25 | 54 | 2.18 |
| acetate | 81 | 65 | 0.81 | 22 | 20 | 0.94 |
| formate | 61 | 27 | 0.45 | 18 | 8 | 0.48 |
| Al | 60 | 180 | 3.01 | 17 | 57 | 3.29 |
| adipate | 46 | 35 | 0.75 | 13 | 10 | 0.79 |
| maleate | 43 | 31 | 0.71 | 12 | 9 | 0.73 |
| $NO_2^-$ | 40 | 69 | 1.72 | 13 | 21 | 1.68 |
| glycolate | 23 | 38 | 1.63 | 7 | 11 | 1.67 |
| Fe | 19 | 48 | 2.55 | 5 | 15 | 2.85 |
| $Br^-$ | 12 | 19 | 1.60 | 4 | 6 | 1.52 |
| $Cl^-:Na^+$ (1.80) | 1.76 | 1.80 | 1.02 | - | - | - |
| $Mg^{2+}:Na^+$ (0.12) | 0.13 | 0.15 | 1.14 | - | - | - |
| $Ca^{2+}:Na^+$ (0.038) | 0.05 | 0.06 | 1.19 | - | - | - |
| $SO_4^{2-}:Cl^-$ (0.14) | 0.35 | 0.36 | 1.02 | - | - | - |
| $MSA:nssSO_4^{2-}$ | 0.19 | 0.20 | 1.08 | - | - | - |



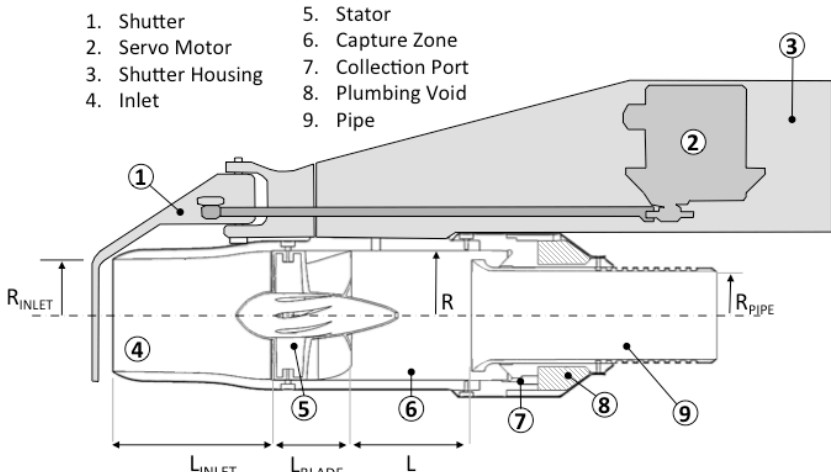

Figure 1: Probe schematic showing main components and key dimensions (see Table 1)





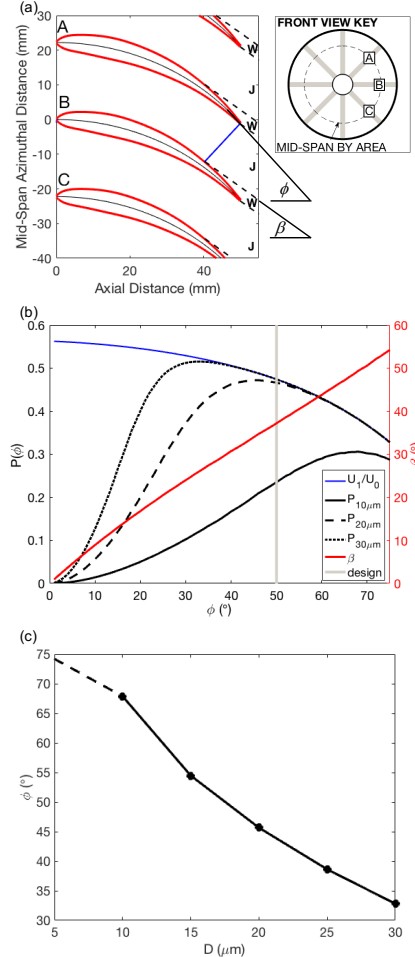

Figure 2: Optimization of stator blade angles. (a) schematic of blade geometry and associated wake (W) and jet (J) structure, (b) stator performance as a function of blade angle (see text for details), and (c) optimum blade angle using method shown in
5    (b) calculated over a range of droplet diameters (dashed line indicates low confidence of applicability of theoretical model).





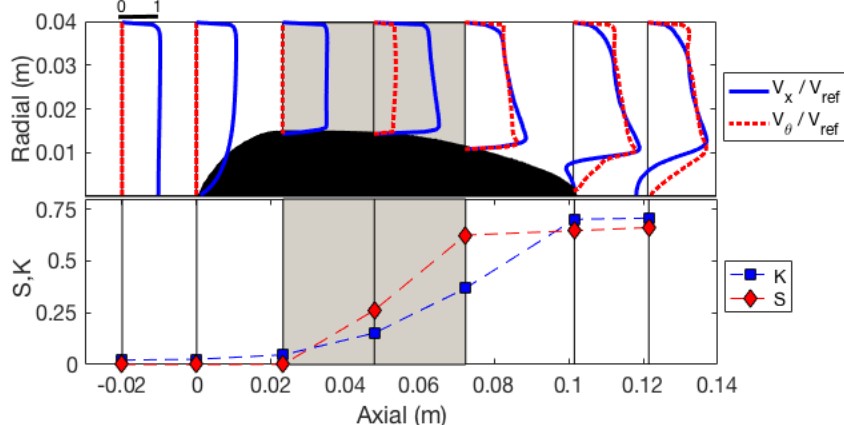

Figure 3: (a) profiles of axial and tangential velocity ratio at seven stations across the stator, and (b) axial dependence of total

loss coefficient (K) and swirl number (S). The shaded region indicates the extent of the blade chord and a scale is indicated

5   above the first profile.





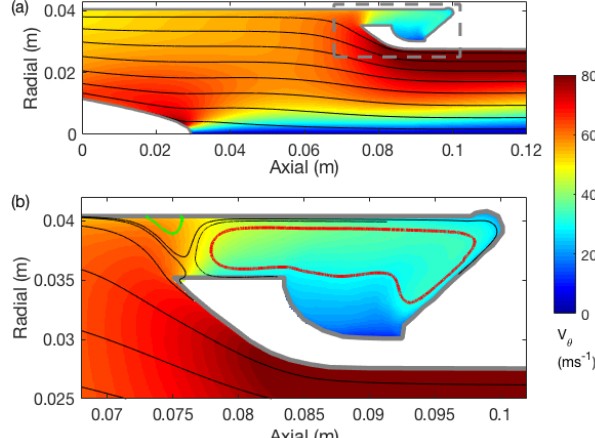

Figure 4: CFD derived azimuthally-averaged streamline pattern downstream of the stator blades with contours of tangential velocity showing (a) an axisymmetric slice of the collection zone, and (b) a close-up of the collection zone bounded by the dashed box in (a). Flow streamlines indicate a region of separation followed by reattachment on the outer wall (green streamline) and a broad recirculation zone (red streamline)



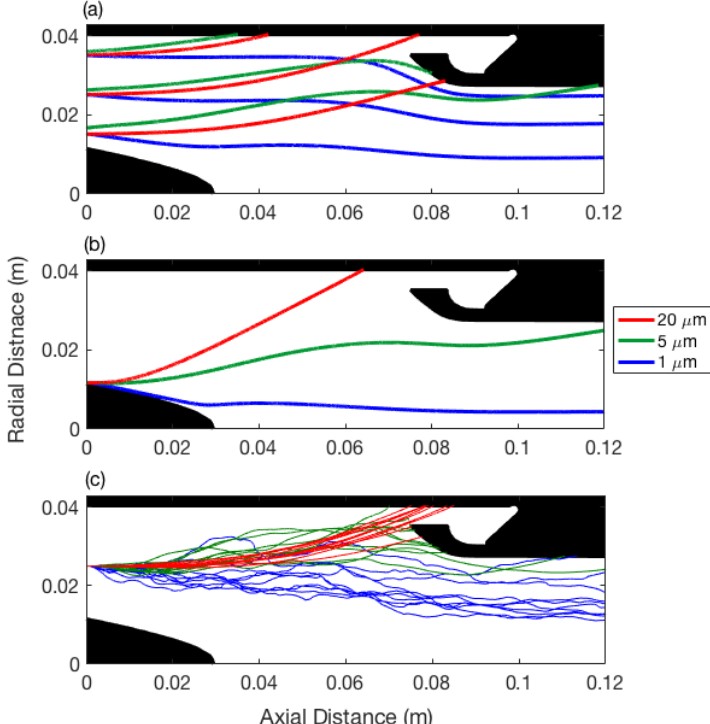

Figure 5: Capture zone droplet trajectories showing a) variation of droplet trajectory with diameter and radial position with droplets in equilibrium with upstream flow, b) trajectories of droplets shed at the blade trailing edge with zero initial momentum, and c) effect of flow turbulence on an ensemble of trajectories, released as per (a) in the middle location.





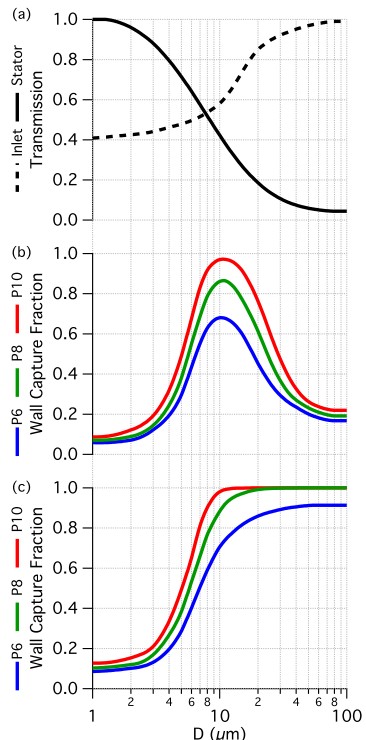

Figure 6: Statistics of trajectory ensembles across a range of droplet diameter. Panel (a) shows the transmission computed for the inlet and stator, which are unaffected by the pipe position. In panels (b) and (c) the fraction of droplets that migrate to the outer walls is compared between three pipe positions for (b) droplets at equilibrium with the upstream flow and (c) droplets shed from the blade trailing edge with zero initial momentum





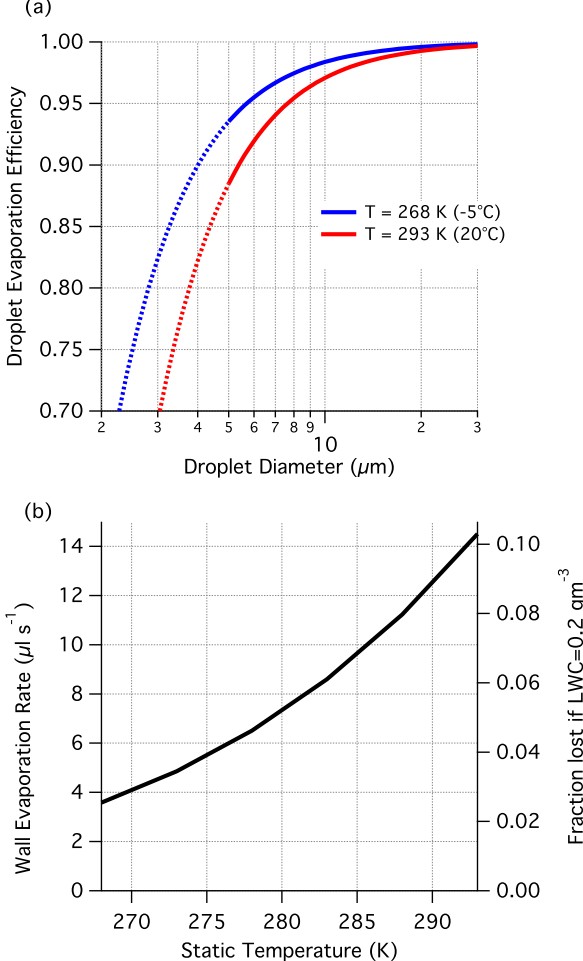

Figure 7: Evaporation losses from (a) droplets migrating to the wall ($\eta_{evap}$), and (b) water on the walls ($\delta_{evap}$). Loss calculations correspond to nominal flight conditions for the C-130 (or equivalent) aircraft with true airspeed 115 ms$^{-1}$ and a static pressure 850 hPa. Interpolation is acceptable for static temperature lying within the bounds shown in (a), and changes in static pressure (within 500-1000 hPa) can be assumed to have negligible impact on evaporation. The transition from solid to dotted curves reflects the lower bound for activated droplets for purposes of the design. In (b), the right axis is shown as an illustration of the effect of wall evaporation in the context of TCE (see text for details).





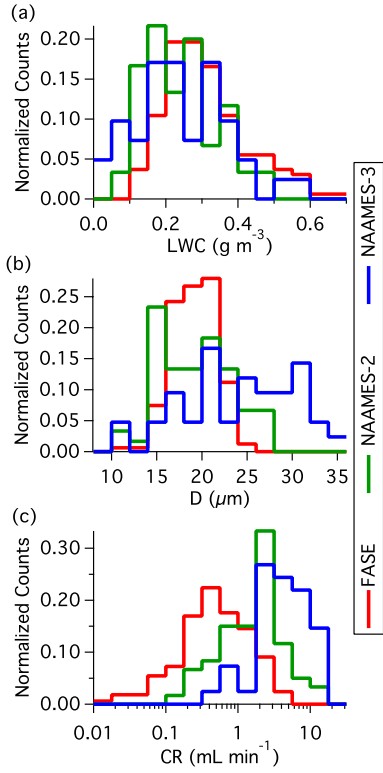

Figure 8: Campaign histograms showing the variability of (a) LWC, (b) effective diameter (D), and (c) collection rate (CR)





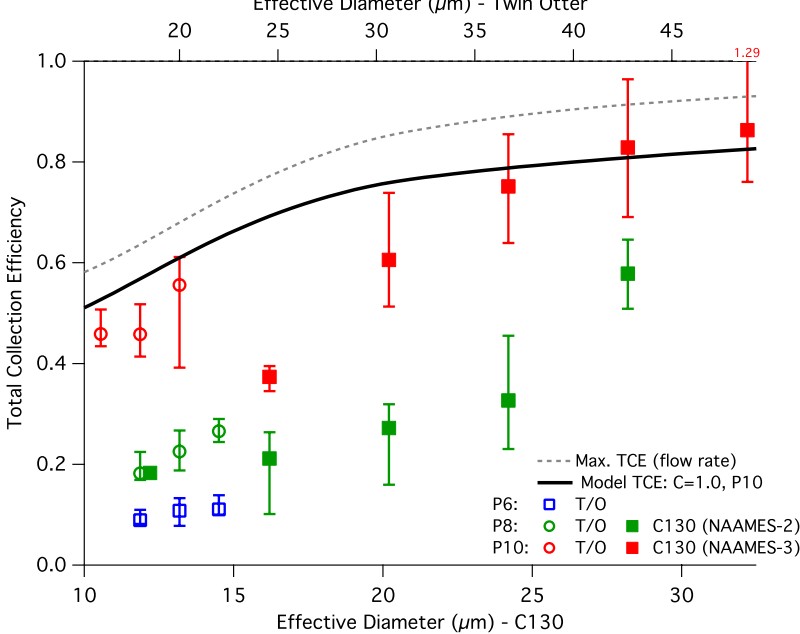

Figure 9: Bulk performance estimate across the NAAMES-2, FASE and NAAMES-3 campaigns. Markers represent median (50%) statistics, while bars represent the 25% to 75% range. Top and bottom axes are scaled for Stokes Number similarity between corresponding airspeeds on the C-130 and Twin Otter (T/O) platforms. Data are grouped into bins of effective diameter based on the range of conditions encountered in each campaign. Model predicted performance is shown for the P10 case together with the model predicted maximum performance based solely on inlet transmission (see text for details).

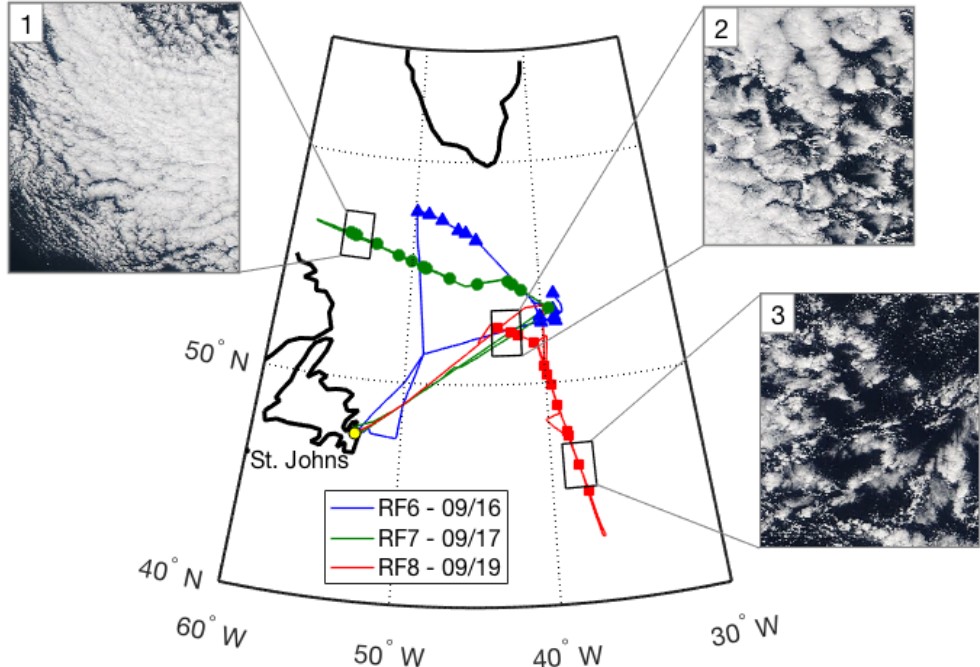

Figure 10: Flight tracks and cloud water sample locations during cold air outbreak case flown during NAAMES-3. Inset
MODIS visible images correspond to selected cloud water cases and highlight the range of cloud conditions





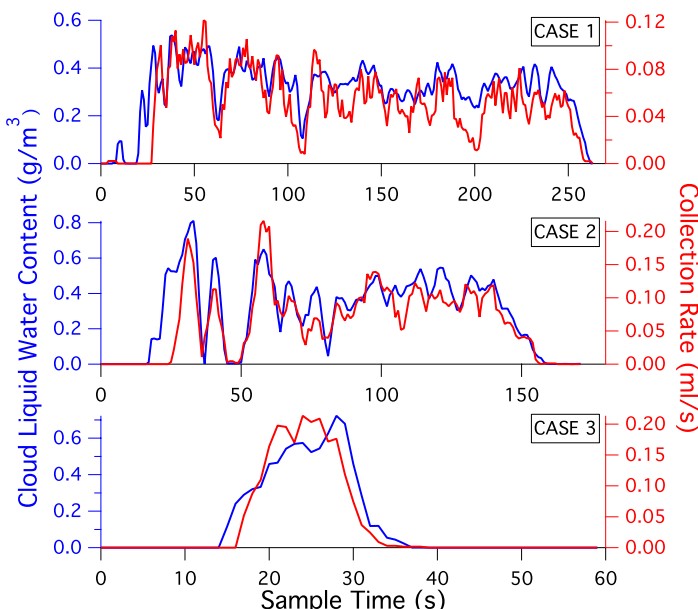

Figure 11: Comparison of instantaneous collection rate (CR) time series with observed cloud liquid water content (LWC) derived from the CDP for the three cases identified in Figure 11.



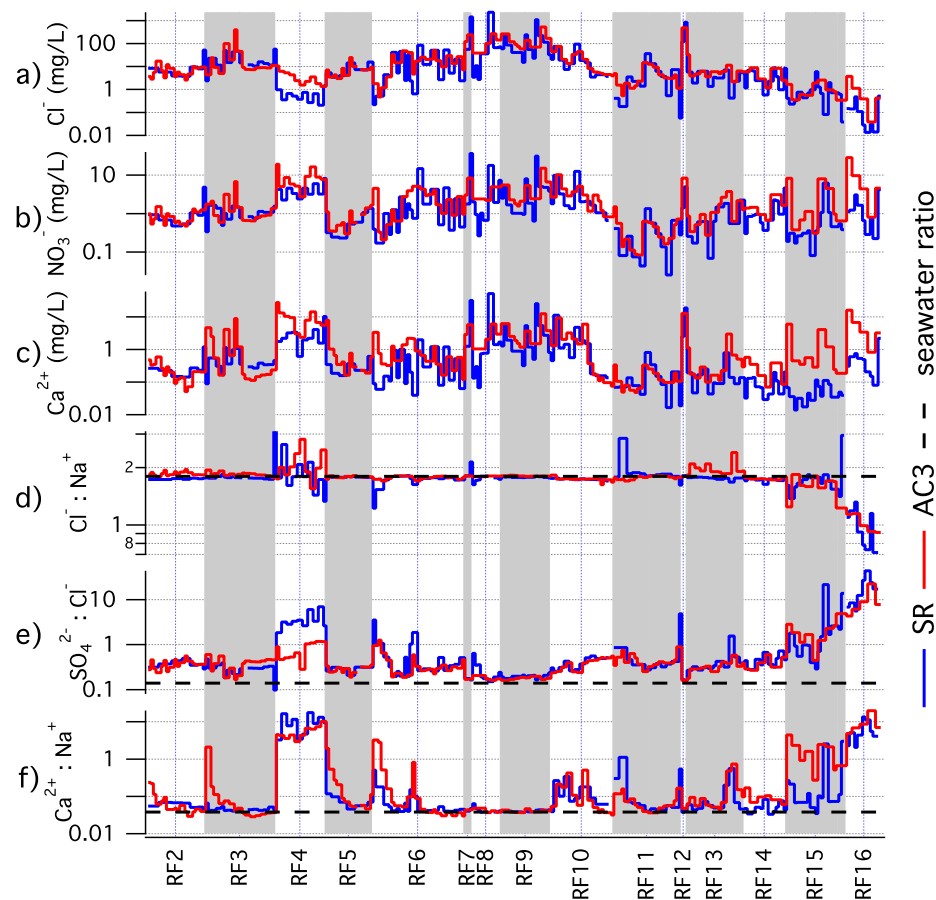

Figure 12: Pseudo-time series of selected chemical species concentration (a-c) and ratios (d-f) observed during the intercomparison between AC3 and the slotted rod (SR) as part of the FASE campaign.