# Peer review of "Development and characterization of a high-efficiency, aircraft-based axial cyclone cloud water collector"

_Atmospheric Measurement Techniques, 2018_

## Referee Comment (RC1) · D. Baumgardner (Referee) · 20 Jul 2018

Having been involved in the development of the first cyclone water collector developed by Straub and Collett, I am delighted to see that the idea has survived and that there is now a newer version that addresses some of the limitations of the first version.

I find little to critique in this manscript as it is very well organized in sufficient detail for the reader to understand virtually every aspect of the design, from the mechanical structure to the airflow characteristics. The modeling that is done is adequate for doing the necessary sensitivity studies that were used to select the stator blade angles, as well as where to position the other critical components for extracting the water.

[Figure]

The questions that I have are not serious enough to delay publication, but perhaps the authors could say a word or two about the following:

1) Unless I somehow missed it, there seems to be no airflow modeling that takes into account the location of the AC3 on the Otter. Given that the AC3 is mounted on a pylon only 12 inches from the fuselage of the C-130, fairly far back from the nose, it is likely it is sampling cloud droplets, some whose concentrations are enhanced with respect to the free stream. Likewise, the mounting on the Twin Otter where the airflow might be less disturbed but possibly still somewhat disturbed. Something should be discussed in this regard in the manuscript.

2) How do attitude angle changes impact collection? I don't think this was mentioned?

3) It is mentioned that the water is pumped into the cabin but this must have only been on the C-130?

4) Figure 2C and the choice of 50 degrees, please amplify why this angle was chosen. A suggestion that in this figure the x axis should be D^3 not D. My reasoning is that both LWC and relaxation time depend on mass not diameter, hence the choice of 50 might be better explained if mass rather diameter is referred to since it is LWC that you are trying to maximize.

5) Are there plans for a wind tunnel study? I strongly recommend seeking funding to put the AC3 in the NASA Glenn wet wind tunnel.

6) Photos of the AC3 on the Twin Otter and C-130 would be very illustrative.

---

## Referee Comment (RC2) · Anonymous Referee #2 · 26 Jul 2018

This is a very clearly written manuscript on a performance of recently modified and improved aircraft water collector designed to capture cloud water samples (separated from aerosol) for post-flight analysis. This major research and engineering effort of the large group of authors resulted in construction of a new, very useful instrument for chemical characterization of cloud water. AMT is a right choice to publish results of this study. The design and theoretical performance of the probe analyzed by means of numerical modeling are carefully described. Then, results of the in-flight measurements on two very different research aircrafts, accompanied by post-flight analysis of collected samples are presented in discussed. The only comment I have to the manuscript is that there is not much on a performance of the probe in the presence of larger droplets,

e.g. drizzle. Was this problem analyzed? Does it influence capture efficiency? Does presence of larger droplets (presumably grown in a different region of the cloud than the measurements are performed) affect the results? I do not ask for an in-depth analysis of the above, but for some comments which might be useful for the future users of the instrument. Additional information on the author contribution along AMT guidelines should also be included. These are my suggestions for a minor revision of the manuscript.

---

## Author Comment (AC1) · 27 Jul 2018

We thank both reviewers for their positive remarks and helpful feedback.

**Author Comment – Reviewer 1**

Having been involved in the development of the first cyclone water collector developed by Straub and Collett, I am delighted to see that the idea has survived and that there is now a newer version that addresses some of the limitations of the first version.

I find little to critique in this manuscript as it is very well organized in sufficient detail for the reader to understand virtually every aspect of the design, from the mechanical structure to the airflow characteristics. The modeling that is done is adequate for doing the necessary sensitivity studies that were used to select the stator blade angles, as well as where to position the other critical components for extracting the water.

The questions that I have are not serious enough to delay publication, but perhaps the authors could say a word or two about the following:

1) Unless I somehow missed it, there seems to be no airflow modeling that takes into account the location of the AC3 on the Otter. Given that the AC3 is mounted on a pylon only 12 inches from the fuselage of the C-130, fairly far back from the nose, it is likely it is sampling cloud droplets, some whose concentrations are enhanced with respect to the free stream. Likewise, the mounting on the Twin Otter where the airflow might be less disturbed but possibly still somewhat disturbed. Something should be discussed in this regard in the manuscript.

Currently we have not done any CFD modelling of the flow around the airframe on either platform, but we accept that this is a valid concern worthy of future investigation. Furthermore, the mounting of the CDP used to provide DSD and LWC information is also subject to potential airframe induced flow changes, which may affect size-dependent concentrations. Although not suitable to quantitatively assess the suitability of the mounting location on the C-130, we can report that the same location has been used on a past mission to fuselage-mount a CDP. While this does not tell us about subtle size-dependent enhancements/reductions, the data collected during those flights does indicate that the DSD is not catastrophically disrupted by the proximity to the fuselage.

Since the mounting location may contribute to differences we found in the collection efficiency comparisons between platforms, we have added text to the manuscript to highlight this.

2) How do attitude angle changes impact collection? I don't think this was mentioned?

In terms of observations, we do not have enough data to currently answer this mainly because we typically flew at the same speed in low clouds and did not systematically get samples at the beginning and end of flight where there could be an aircraft weight difference. This is an important aspect to consider though, which we will continue to monitor as more data is collected. The modelling framework was not set up to answer this question, because we used an axisymmetric model for the inlet. Instinctively, we may expect the effect to be minimal after the inlet plane, since we have shown that surface impaction is unlikely to cause significant water

loss. We have added a line in the text to recognize that pitch and yaw angles may affect performance.

3) It is mentioned that the water is pumped into the cabin but this must have only been on the C-130?

The same system was used on the Twin Otter. The same PFA tubing was run up the pylon along the wing and the collection system was mounted in a rack just below the wing root. The tubing length was quite similar to the C-130 despite the distance along the wing.

4) Figure 2C and the choice of 50 degrees, please amplify why this angle was chosen. A suggestion that in this figure the x axis should be D3 not D. My reasoning is that both LWC and relaxation time depend on mass not diameter, hence the choice of 50 might be better explained if mass rather diameter is referred to since it is LWC that you are trying to maximize.

We agree that the rationale for 50 degrees needs more weight and have added text to resolve this.

Figure 2C shows the optimal angle given different DSDs. As mentioned in the text, for these reduced-order analyses we use a log-normal mass distribution centered at D. So, the optimization already accounts for the fact that we want to maximize mass. The fact that we chose 50 degrees, was driven by the fact that we anticipated that most of the cloud water (on a mass basis) was between 15-20 microns, in the types of clouds that we anticipated sampling. Figure 2C tells us that had we optimized for DSDs with smaller droplets we may have wanted a greater turning angle, while DSDs centered at larger sizes may have warranted a lesser angle. Hopefully the revised text helps to convey this better, but we believe that the figure does not need to be changed.

5) Are there plans for a wind tunnel study? I strongly recommend seeking funding to put the AC3 in the NASA Glenn wet wind tunnel.

This is something that has been discussed over the last couple of years. We hope that this is something that could materialize in the future.

6) Photos of the AC3 on the Twin Otter and C-130 would be very illustrative.

We do not have appropriate photos for this purpose.

**Author Comment - Reviewer 2**

This is a very clearly written manuscript on a performance of recently modified and improved aircraft water collector designed to capture cloud water samples (separated from aerosol) for post-flight analysis. This major research and engineering effort of the large group of authors resulted in construction of a new, very useful instrument for chemical characterization of cloud water. AMT is a right choice to publish results of this study. The design and theoretical performance of the probe analyzed by means of numerical modeling are carefully described. Then, results of the in-flight measurements on two very different research aircrafts, accompanied by post-flight analysis of collected samples are presented in discussed. The only comment I have to the manuscript is that there is not much on a performance of the probe in the presence of larger droplets, e.g. drizzle. Was this problem analyzed? Does it influence capture efficiency?

If the LWC of drops exceeding the 50 micron upper bound of the CDP were a substantial fraction of the total LWC, then our analysis of collection efficiency would be affected. This is stated in the text on p18 L8-10.

In general, our analysis suggests large drops are collected with high efficiency but we do not have a way of isolating the observed collection efficiency contribution from the drizzle, because we do not currently have enough data points to statistically isolate drizzle contributions from changes in effective diameter. Our expectation would be that drizzle is collected at between 85%-90% efficiency. Drizzle drops may experience greater shatter losses because of their larger Weber Number, but as already suggested in the text, the data do not indicate that shatter related losses are substantial.

Does presence of larger droplets (presumably grown in a different region of the cloud than the measurements are performed) affect the results?

Yes, this is a limitation with making a bulk measurement. We reference Bator and Collett, 1997 in the text, which documents composition dependence on size. We expect, based on Fig. 9, that we get quite a flat collection efficiency above 25 microns (at C-130 speeds), so it seems that large drizzle drops would not skew the bulk composition when the effective diameter is >25 microns, since we would expect them to be collected at approximately the same efficiency. Drizzle may skew the bulk composition when sampling in small droplet environments such as cloud base or in polluted clouds, because those drops would be preferentially sampled.

I do not ask for an in-depth analysis of the above, but for some comments which might be useful for the future users of the instrument. Additional information on the author contribution along AMT guidelines should also be included.

**Added, thanks.**

These are my suggestions for a minor revision of the manuscript.